# MonoScale: Scaling Multi-Agent System with Monotonic Improvement

**Shuai Shao** [* 1]  **Yixiang Liu** [* 1]  **Bingwei Lu** [1]  **Weinan Zhang** [1 2]

## Abstract

In recent years, LLM-based multi-agent systems (MAS) have advanced rapidly, using a router to decompose tasks and delegate subtasks to specialized agents. A natural way to expand capability is to **scale up the agent pool** by continually integrating new functional agents or tool interfaces, but naive expansion can trigger **performance collapse** when the router cold-starts on newly added, heterogeneous, and unreliable agents. We propose **MonoScale**, an expansion-aware update framework that proactively generates a small set of agent-conditioned familiarization tasks, harvests evidence from both successful and failed interactions, and distills it into auditable natural-language memory to guide future routing. We formalize sequential augmentation as a contextual bandit and perform trust-region memory updates, yielding a monotonic non-decreasing performance guarantee across onboarding rounds under a non-interfering expansion assumption. Experiments on GAIA and Humanity's Last Exam show stable gains as the agent pool grows, outperforming naive scale-up and strong-router fixed-pool baselines. Our code is available here.

## 1. Introduction

Multi-agent systems (MAS) built on large language models (LLMs) are emerging as a powerful paradigm for solving complex tasks (Chen et al., 2025; Yang et al., 2024; Guo et al., 2024; Liao et al., 2026). Unlike a single model that produces an answer end-to-end, an MAS typically includes a router that decomposes an input problem, dispatches subtasks to specialized agents (Hu et al., 2025; Yang et al., 2025b; Yue et al., 2025)(e.g., retrieval, code execution, planning, tool use, or domain experts), and then aggregates

their outputs into a final response. This modular "divide–and–conquer" structure enables flexible collaboration and makes MAS particularly effective for multi-step workflows, tool-augmented reasoning, and long-horizon task execution.

In real deployments, MAS are often continuously evolving: new tools, plugins, external APIs, or specialized agents may be integrated over time, leading to a dynamically expanding agent pool (Kim et al., 2025; Microsoft, 2025). While adding a stronger agent or one that covers new skills should, in principle, increase the system's capability ceiling, naive scale-up in practice can yield the opposite outcome—overall performance may stagnate, degrade, or even exhibit a clear *performance collapse*. This failure mode is not merely hypothetical: Figure 1 shows that, on GAIA under cold-start routers, increasing the agent pool size can hurt end-to-end performance (e.g., DeepSeek-V3.2 drops from 0.558 at 5 agents to 0.491 at 10; Qwen3 peaks at 0.461 with 7 agents but falls to 0.424 at 10). A key reason is the router's *cold start*: once a new agent $a_{new}$ becomes available, the router typically lacks grounded knowledge of what the agent is good at, when it is reliable, what its boundaries and failure modes are, or how to interact with its tools (e.g., interfaces, output formats, and frequent errors). If the router assigns tasks to $a_{new}$ too early or too aggressively, mis-routing decisions can be amplified at the system level and reduce end-to-end reward—for example, sending high-precision requests to an unstable retrieval agent, delegating complex tool calls to an agent unfamiliar with the interface, or introducing a brittle link in a multi-step workflow that causes the entire chain to fail.

Existing approaches for improving multi-agent coordination broadly fall into two lines, yet neither fully addresses the central challenge of *open-ended* MAS scaling. The first line focuses on a **static agent pool**, assuming that the agent set and its capability profile remain stable between training and deployment (Yue et al., 2025; Zhang et al., 2024; Liu et al., 2025b); the research emphasis is then on optimizing routing policies, improving division of labor, or planning more efficiently within a fixed pool. While effective in static settings, these methods typically lack a controlled update mechanism that can *prevent performance regressions* when new agents are continually added, and they do not provide a unified formalization for *online updating under continual agent augmentation* in open-ended systems. In parallel, some

---

[*]Equal contribution [1]Shanghai Jiao Tong University, Shanghai, China [2]Shanghai Innovation Institute, Shanghai, China. Correspondence to: Weinan Zhang <wnzhang@sjtu.edu.cn>.

*Proceedings of the 43$^{rd}$ International Conference on Machine Learning*, Seoul, South Korea. PMLR 306, 2026. Copyright 2026 by the author(s).

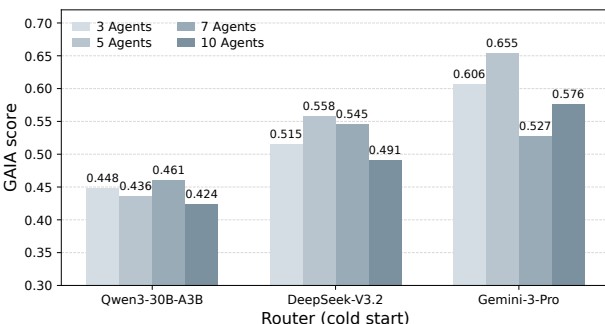

*Figure 1.* Motivation for **MonoScale**: without an expansion-aware familiarization and update mechanism, routers cold-start on newly added agents, causing unstable routing decisions and potential performance collapse during scale-up. Bars show GAIA scores for different agent pool sizes ($N \in \{3, 5, 7, 10\}$) under cold-start routers (Qwen3-30B-A3B, DeepSeek-V3.2, and Gemini-3-Pro).

recent work has started to examine **dynamic expansion** in LLM routing (Wang et al., 2025; Zhang et al., 2025c), where the system routes across an enlarging set of available experts over time. However, scaling an MAS is substantially harder: newly integrated *agents* are far more heterogeneous (tools, memory, skills, roles, and interfaces), and their quality can be highly uneven in practice (brittle tool calls, API failures, retrieval noise, or unstable behaviors). As a result, the router faces a sharper **cold start** problem, and a fixed onboarding suite to "trial-run" the new agent is often insufficient: (i) it cannot be adapted to each agent's unique boundaries and failure modes; and (ii) it may under-cover the true deployment distribution, creating a risk of agents that appear usable during onboarding but still collapse after real deployment. In short, what dynamic MAS scaling truly requires is an **adaptive** (agent-conditioned) and **conservative** update protocol, so that end-to-end performance can *improve stably* as the agent pool grows.

To bridge this gap, we propose **an update framework for sequentially expanding open-ended MAS**. Whenever a new agent is introduced, rather than waiting for real user traffic to surface failures, we proactively customize a small set of **familiarization tasks** tailored to the new agent and use them to collect controlled feedback. Our mechanism has two stages: (i) **active familiarization via task customization**, where tasks are synthesized to probe the new agent's likely strengths, interface constraints, and failure modes, enabling the router to learn *when to use / when not to use* the agent; and (ii) **router update via memory**, where we update the router's policy by maintaining an auditable text memory distilled from experience. Concretely, the memory captures not only patterns behind *successful* orchestration, but also lessons from *failed* worker calls and mis-routings; we abstract these signals into a compact set of actionable principles (e.g., capability boundaries, safe/unsafe contexts, interface caveats, and common failure patterns) that guide

future routing decisions. By continuously learning these routing principles as the agent pool evolves, the router becomes less prone to brittle dispatching caused by unfamiliarity with newly integrated agents, enabling end-to-end performance to scale more stably with additional agents. On the theory side, we cast orchestration under sequential augmentation as a contextual bandit and prove that, under a non-interfering expansion assumption, our trust-region memory optimization yields a monotonic non-decreasing performance guarantee across onboarding rounds.

We systematically evaluate the sequential expansion process of open multi-agent systems on the GAIA (Mialon et al., 2023) validation set and the multiple-choice subset of Humanity's Last Exam (Phan et al., 2025). We compare: (i) naive scale-up: using a Qwen-3-30B-A3B-Instruct router, which simply adds more agents without task customization or memory updates; results show that performance not only fails to improve, but instead exhibits a clear collapse as the number of agents continues to grow; (ii) multi-agent systems with SOTA models as routers (e.g., GPT-5); and (iii) our expansion-aware protocol, which includes customized familiarization tasks and natural-language memory updates. Our results demonstrate that this approach substantially reduces the risk of post-expansion performance collapse and yields more stable, sustained improvements across multiple onboarding rounds. Notably, with task customization and memory, the Qwen-3-30B-A3B-Instruct router benefits from scaling up the agent pool and can even outperform the "strong-router + fixed agent pool" baseline, indicating that adaptive router updates—rather than backbone strength alone—are crucial for robust MAS scaling. In addition, we further evaluate robustness under a noisy agent pool setting, i.e., the pool contains malfunctioning agents, and compare our method against strong-router baselines: even the strongest model (Gemini-3-pro) can suffer a performance collapse when faced with a noise-heavy agent pool, whereas our method remains stable and maintains MAS performance even under such noise.

**Contributions.** Our main contributions are three-fold:

- **Problem formalization:** we formally define MAS scaling under sequential expansion, and introduce a unified model and notation for analyzing cold-start-induced performance collapse.

- **Method and protocol:** we propose an expansion-aware task customization (data synthesis) strategy and a router update protocol that actively elicits evidence about each newly added agent's profile, and updates the router via auditable, controllable, and rollbackable natural-language memory editing.

- **Theoretical guarantee:** under a contextual bandit formulation, we prove that, under a non-interfering expan-

sion assumption, our trust-region memory optimization yields a monotonic non-decreasing performance guarantee during scaling, providing a verifiable conditional lower-bound for open MAS expansion.

## 2. Related Works

**LLM-based multi-agent systems and orchestration.** LLM-based multi-agent systems (MAS) (Yang et al., 2025b; Chen et al., 2025; Yang et al., 2024) typically rely on a central router or coordinator to decompose tasks (Hong et al., 2024; Li et al., 2023; Wu et al., 2023), dispatch subtasks to specialized agents (Qian et al., 2024; Hu et al., 2025) (e.g., retrieval, tool use, code execution), and aggregate results (Wang et al., 2023; 2024), enabling tool-augmented and long-horizon workflows. In this setting, a large body of work studies how to improve planning, division of labor, and routing decisions under a static agent pool (Yue et al., 2025; Zhang et al., 2024; Liu et al., 2025b), *i.e.,* the agent set and its capability profile are assumed fixed between training and deployment. Such approaches are effective when the action space is stable, but they usually do not provide a controlled *online updating under continual agent augmentation* mechanism that prevents regressions once the agent pool changes over time.

**Dynamic MAS evolution and the cold-start problem.** Recent work explores self-evolving LLM-based multi-agent systems, where teams, roles, or interaction topologies are generated, optimized, or reorganized automatically (Yang et al., 2025b; Zhuge et al., 2024; Zhang et al., 2025a; Lu et al., 2025). Such internal adaptation can improve a MAS, but unconstrained self-evolution may drift into emergent risks (Shao et al., 2026), so the evolutionary capability and reliability of agents must be explicitly evaluated rather than assumed (Fu et al., 2026; Yang et al., 2025a). Crucially, these approaches also do not address dynamic scale-up—continuously integrating new specialist or generalist agents into an existing system. A related line studies system-level expansion by routing over growing pools of heterogeneous models (Wang et al., 2025; Zhang et al., 2025c): UniRoute routes queries to previously unseen LLMs via learned model embeddings (Jitkrittum et al., 2025), and StageRoute performs online deployment and routing over a streaming model inventory under capacity and cost budgets (Li & Li, 2026). However, these address *model-level* routing, where the decision unit is a single standalone LLM. Agent routing in MAS is harder: agents are far more heterogeneous (tools, memory, skills, roles) and less reliable in practice (tool/API failures, brittle interfaces, retrieval noise), so routing must jointly weigh capability and reliability and produce a multi-step orchestration plan rather than a single-model choice. Closest to our setting, EvoRoute performs experience-driven self-routing within an agentic system (Zhang et al., 2026a),

but targets the cost, latency, and accuracy trade-off rather than preventing cold-start collapse during agent-pool expansion. These methods are thus complementary to MonoScale rather than directly applicable baselines.

**Agent learning.** A large body of work improves LLM-based agents via diverse forms of learning. One direction maintains explicit memory modules that distill long-term experience—trajectories, tool-use outcomes, and failure cases—for reuse in later decisions (Zhang et al., 2026c; Zhou et al., 2025; Zhang et al., 2025d; Ouyang et al., 2025; Wang et al., 2026). Another line applies reinforcement learning and its variants to improve agent behavior (Shao et al., 2024; Su et al., 2026; Dong et al., 2025; Liao et al., 2025; Zhang et al., 2025b); recent *training-free* variants such as Training-Free GRPO (Cai et al., 2025) move optimization from weight updates to inference-time, experience-conditioned policy adjustments. A further line distills modular, reusable *skills* from experience that can be retrieved or composed for new tasks (Zhang et al., 2026b; Xia et al., 2026; Lu et al., 2026). More broadly, classic ideas on conservative, stable policy updates remain relevant for agent learning (Schulman et al., 2017; Vieillard et al., 2020; Laroche et al., 2019). We model MAS orchestration as a contextual bandit and update the router via memory during expansion, mitigating performance collapse from agent augmentation while preserving existing capabilities.

## 3. Scaling Multi-Agent System with Monotonic Improvement

In this section, we study the sequential augmentation of open-ended Multi-Agent Systems (MAS) and present an expansion-aware router update framework. We first formalize orchestration under an evolving agent pool and articulate the monotonicity challenge induced by the expanding action space. We then develop a memory-based policy improvement procedure with conservative updates, and defer the algorithmic instantiation and theoretical analysis to the subsequent subsections.

### 3.1. Preliminaries

#### 3.1.1. CONTEXTUAL BANDIT AT EXPANSION STAGE $k$

We model routing in an evolving multi-agent system as a sequence of contextual bandit problems indexed by the expansion stage $k \in \{0, 1, 2, \dots\}$. At stage $k$, the system has an agent pool $\mathcal{S}_k$, which induces an action space $\mathcal{Y}_k := \mathcal{Y}(\mathcal{S}_k)$ of feasible orchestration plans (routing schemes).

In each interaction round at stage $k$: (i) a task context $x \sim \mathcal{D}$ is drawn i.i.d. from a fixed deployment distribution,[1] (ii) the

---

[1]We assume the task distribution is stationary across expan-

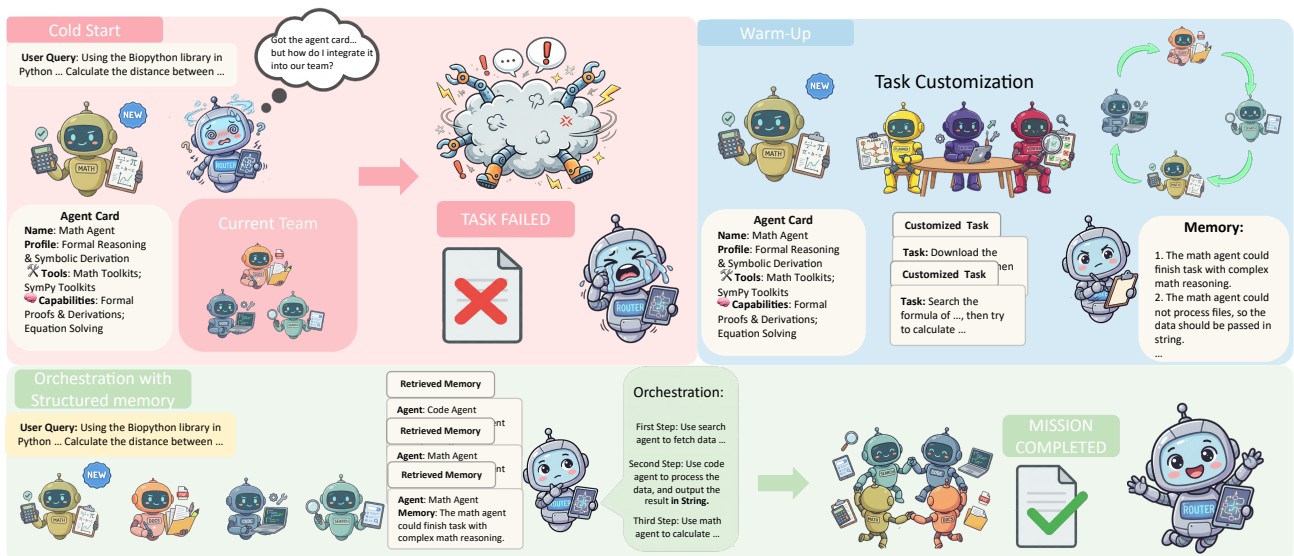

*Figure 2.* Overview of our expansion-aware familiarization-and-memory-update protocol. After adding a new agent, we generate customized warm-up tasks (conditioned on agent cards), collect both success and failure traces under the current router, distill the evidence into structured routing principles (memory candidates), and select a safe memory update with a conservative fallback to prevent brittle mis-routing during scale-up.

router selects an orchestration plan $y \sim \pi(\cdot \mid x)$ with $y \in \mathcal{Y}_k$, (iii) executing $y$ with agent pool $\mathcal{S}_k$ yields a *verifiable* reward $r_k(x, y) \in [0, 1]$, as determined by an automatic evaluator. The stage-$k$ objective is

$$J_k(\pi) := \mathbb{E}_{x \sim \mathcal{D}} \, \mathbb{E}_{y \sim \pi(\cdot \mid x)} \big[ r_k(x, y) \big]. \quad (1)$$

### 3.1.2. ROUTER POLICY WITH EDITABLE MEMORY

The MAS is coordinated by a central **Router** that maps a task $x$ to an orchestration plan $y$. We adopt a parameter-efficient setup where the router is a frozen LLM whose behavior is modulated by an editable memory $m$ (a text buffer or a list of text entries) under a token-length budget:

$$m \in \mathcal{M}, \qquad \text{len}(m) \leq K. \quad (2)$$

Conditioning the frozen LLM on the task $x$, the memory $m$, and the descriptions of the currently available agents (e.g., agent cards for $\mathcal{S}_k$) induces a stochastic routing policy

$$\pi_m^k(y \mid x) := p_{\text{LLM}}(y \mid x, m; \mathcal{S}_k), \qquad y \in \mathcal{Y}_k. \quad (3)$$

Thus, at each stage $k$, policy optimization reduces to selecting a memory state $m$ within $\mathcal{M}$.

### 3.1.3. SEQUENTIAL AUGMENTATION AND CONSERVATIVE EMBEDDING

We consider **sequential augmentation**: starting from an initial agent pool $\mathcal{S}_0$, each expansion step adds a new agent

sions; the expansion only changes the available agent pool and thus the feasible routings and their outcomes.

$a_k$,

$$\mathcal{S}_k = \mathcal{S}_{k-1} \cup \{a_k\}, \qquad \mathcal{Y}_{k-1} \subseteq \mathcal{Y}_k, \quad (4)$$

where the nesting reflects that any plan that only calls agents in $\mathcal{S}_{k-1}$ remains a feasible plan after adding $a_k$.

A naive update after expansion may degrade performance due to mis-routing to the unfamiliar new agent. Our goal is **monotonic improvement across expansions**, i.e.,

$$J_k(\pi_k) \geq J_{k-1}(\pi_{k-1}), \qquad \forall k \geq 1, \quad (5)$$

where $\pi_k$ denotes the deployed router policy at stage $k$.

To make (5) well-defined despite the action-space change, we introduce a conservative embedding of the pre-expansion policy into the post-expansion action space. Given any policy $\pi_{k-1}$ defined on $\mathcal{Y}_{k-1}$, define its *conservative lift* $\pi_{k-1}^{\uparrow}$ on $\mathcal{Y}_k$ by

$$\pi_{k-1}^{\uparrow}(y \mid x) := \begin{cases} \pi_{k-1}(y \mid x), & y \in \mathcal{Y}_{k-1}, \\ 0, & y \in \mathcal{Y}_k \setminus \mathcal{Y}_{k-1}. \end{cases} \quad (6)$$

**Backward-compatible expansion.** We assume expansions are *non-interfering*: adding a new agent only enlarges the set of available routing options, while leaving the execution and evaluation of any plan that does not invoke the new agent unchanged. Under this assumption, the lifted policy preserves performance:

$$J_k(\pi_{k-1}^{\uparrow}) = J_{k-1}(\pi_{k-1}). \quad (7)$$

Therefore, it suffices to ensure the post-expansion router improves upon the conservative baseline at the new stage,

$$J_k(\pi_k) \geq J_k(\pi_{k-1}^{\uparrow}), \tag{8}$$

which directly implies the cross-expansion monotonicity goal (5). We assume access to a conservative fallback that routes only among previously available agents (i.e., actions in $\mathcal{Y}_{k-1}$). In practice, this is implemented by disabling the newly added agent in the router interface (e.g., omitting an agent for any task provided).

### 3.2. Expansion-Aware Familiarization and Memory Update

As illustrated in Figure 2, each expansion step consists of two stages: *expansion-aware task synthesis* and an *evidence-driven memory update*. When a new agent $a_k$ is integrated, we first synthesize warm-up tasks *conditioned on the new agent's profile* to probe both strengths and failure modes, then execute these tasks under the current router to collect success/failure evidence, and finally distill the evidence into auditable routing principles as structured memory candidates, from which we select a safe update (with a conservative fallback) to mitigate cold-start mis-routing and support stable scaling as the agent pool grows.

#### 3.2.1. EXPANSION-AWARE TASK SYNTHESIS

Our design draws on recent work on open-world agent trajectory synthesis (Guo et al., 2025; Fang et al., 2025; Shi et al., 2025), but adapts the philosophy from benchmark construction to generating a warm-up task distribution centered on new agents for safe Router updates.

For each worker agent in the system, including newly added agents, we maintain a concise *agent card* that summarizes its functional capabilities, available tools, and known behavioral characteristics. When a new agent is introduced, its agent card is provided jointly with the agent cards of existing workers as conditional inputs to a dedicated task synthesizer. The synthesizer itself follows a step-level *planner–executor–validator* loop. Specifically, the planner proposes candidate task specifications, the executor then attempts to carry out the proposed task under the current multi-agent configuration, producing a complete multi-agent interaction process; finally, the validator examines the execution outcome to ensure that the task admits a deterministic solution, follows a coherent execution path, and effectively exposes the capability boundaries or potential mismatch behaviors of the new agent. Only tasks that pass this validation procedure are retained as interaction samples for the subsequent Router familiarization stage.

Following this procedure, after the $k$-th system expansion we obtain a set of expansion-aware synthesized tasks, denoted as $\mathcal{T}_k^{\text{syn}}$, which induces a corresponding warm-up task

distribution $\mathcal{D}_k^{\text{warm}}$. These synthesized tasks are not used to train or update individual worker agents. Instead, they are used to drive the Router, under a fixed policy, to perform batch interactions and sampling. The execution outcomes of all synthesized tasks—including both successful and failed routing cases—are aggregated and subsequently used to support memory updates and policy improvement modeling of the Router in the following stage.

#### 3.2.2. MEMORY UPDATE

During the familiarization stage, the Router interacts with tasks sampled from the warm-up distribution under the current memory state $m_t$: for $x \sim \mathcal{D}_k^{\text{warm}}$, it samples an orchestration plan $y \sim \pi_{m_t}(\cdot \mid x)$, executes it, and receives a scalar reward $r(x, y)$. We design $\mathcal{D}_k^{\text{warm}}$ to closely match the deployment task distribution $\mathcal{D}$ in task form and execution constraints, while amplifying contexts that involve the new agent. Consequently, we treat $\mathcal{D}_k^{\text{warm}}$ as a practical proxy of $\mathcal{D}$ and estimate the expectations in our objective using samples from $\mathcal{D}_k^{\text{warm}}$.

The resulting evidence set (including both successful and failed routing cases) is aggregated to propose candidate memory edits. Concretely, we summarize the warm-up interactions into an experience buffer $\mathcal{B}_k = \{(x_i, y_i, r_i)\}_{i=1}^{N_k}$, and distill auditable routing principles from $\mathcal{B}_k$ to form a set of candidate memory states (see full schema in Appendix E). Let $\tilde{\mathcal{U}}_k(m_t) \subseteq \mathcal{M}$ denote the *evidence-induced* candidate set produced from $\mathcal{B}_k$. During candidate construction, we enforce a *semantic* trust region: newly distilled entries must be compatible with existing routing principles (e.g., via explicit applicability conditions, priorities, or exception clauses). Candidates with irreconcilable conflicts are rejected and thus do not enter $\tilde{\mathcal{U}}_k(m_t)$. Note that $\tilde{\mathcal{U}}_k(m_t)$ can be empty in degenerate cases where no consistent candidate can be distilled from the evidence.

To ensure the update step always has a safe option, we always include a *conservative fallback* in the candidate set. Concretely, let $m_t^{\uparrow}$ denote the memory state obtained by augmenting $m_t$ with a hard prompt constraint that forbids invoking the newly added agent $a_k$ (e.g., "do not call agent $a_k$"). This constraint restricts the router to plans in $\mathcal{Y}_{k-1}$ and thus implements the lifted policy $\pi_{k-1}^{\uparrow}$. We then define the final feasible update set as

$$\mathcal{U}_k(m_t) := \tilde{\mathcal{U}}_k(m_t) \cup \{m_t^{\uparrow}\}, \tag{9}$$

so the optimization can always fall back to the pre-expansion behavior during familiarization.

**Trust-region surrogate objective.** We then directly adopt the surrogate objective of TRPO to evaluate candidate memories. In the contextual bandit setting at stage $k$, given any baseline memory $m_0$, we evaluate a candidate memory $m$

using

$$\mathcal{L}_{m_0}^k(m) := J_k(\pi_{m_0}^k) + \mathbb{E}_{x \sim \mathcal{D}} \Big[ \sum_{y \in \mathcal{Y}_k} \pi_m^k(y|x) \, A_{m_0}^k(x,y) \Big],$$
(10)

We then take the conservative fallback $m_t^\uparrow$ as the baseline and perform the constrained memory update:

$$m_{t+1} \in \arg \max_{m \in \mathcal{U}_k(m_t)} \mathcal{L}_{m_t^\uparrow}^k(m) \quad \text{s.t.} \quad \bar{D}_{\mathrm{KL}}\big(\pi_{m_t^\uparrow}^k \,\|\, \pi_m^k\big) \leq \delta.$$
(11)

Intuitively, memory edits that strongly contradict the current routing principles tend to induce large behavioral shifts and are rejected by the semantic consistency check, so they never enter $\tilde{\mathcal{U}}_k(m_t)$. If no non-fallback candidate satisfies the trust-region constraint, the optimizer selects the conservative fallback, yielding $m_{t+1} = m_t^\uparrow$ (a conservative no-improvement step). We summarize the full implementation of this update—warm-up task synthesis, candidate-memory distillation, and the conservative selection above—in Algorithm 1 (Appendix B).

# 4. Theoretical Analysis

In this section, we provide a monotonic non-decreasing performance guarantee *across expansion stages* for our memory update protocol under the contextual bandit formulation and a non-interfering expansion assumption (Assumption C.1).

## 4.1. Exact Surrogate Objective in Contextual Bandits

Fix an expansion stage $k$ and a baseline memory $m_0$, which induces a routing policy $\pi_{m_0}^k$ over $\mathcal{Y}_k$ and reward $r_k$. Define the baseline value and advantage:

$$V_{m_0}^k(x) := \mathbb{E}_{y \sim \pi_{m_0}^k(\cdot|x)}\big[r_k(x,y)\big], \tag{12}$$

$$A_{m_0}^k(x,y) := r_k(x,y) - V_{m_0}^k(x). \tag{13}$$

**Lemma 4.1** (Exact Bandit Surrogate). *For any memory* $m \in \mathcal{M}$, *the stage-k performance satisfies*

$$J_k(\pi_m^k) = J_k(\pi_{m_0}^k) + \mathbb{E}_{x \sim \mathcal{D}} \Big[ \sum_{y \in \mathcal{Y}_k} \pi_m^k(y \mid x) \, A_{m_0}^k(x,y) \Big]$$

$$=: \mathcal{L}_{m_0}^k(m).$$
(14)

*Proof.* See Appendix C.3. □

## 4.2. Monotonic Improvement Across Expansions

Let $m_{k-1}$ be the deployed memory at stage $k-1$, inducing policy $\pi_{m_{k-1}}^{k-1}$ on $\mathcal{Y}_{k-1}$. After adding a new agent at stage $k$, we consider the conservative lift $\pi_{m_{k-1}}^\uparrow$ on $\mathcal{Y}_k$ (cf. (6)), and realize it via a conservative fallback memory $m_{k-1}^\uparrow$ (e.g., by a prompt constraint that forbids invoking the new agent).

**Lemma 4.2** (Expansion Bridge). *Under the non-interfering expansion assumption (Assumption C.1), the conservative lift preserves performance:*

$$J_k(\pi_{m_{k-1}}^\uparrow) = J_{k-1}(\pi_{m_{k-1}}^{k-1}). \tag{15}$$

*Proof.* See Appendix C.4. □

We update memory at stage $k$ via the trust-region objective

$$m_k \in \arg \max_{m \in \mathcal{U}_k(m_{k-1})} \mathcal{L}_{m_{k-1}^\uparrow}^k(m)$$
$$\text{s.t.} \quad \bar{D}_{\mathrm{KL}}\big(\pi_{m_{k-1}^\uparrow}^k \,\|\, \pi_m^k\big) \leq \delta.$$
(16)

where the feasible edit set is constructed to always include the fallback option $m_{k-1}^\uparrow \in \mathcal{U}_k(m_{k-1})$.

**Theorem 4.3** (Monotonicity Across Expansions). *Under Assumption C.1 (non-interfering expansion) and Assumption C.2 (feasible conservative fallback), let* $\{m_k\}_{k \geq 0}$ *be produced by* (16). *Then for all* $k \geq 1$,

$$J_k(\pi_{m_k}^k) \geq J_{k-1}(\pi_{m_{k-1}}^{k-1}). \tag{17}$$

*In particular,* $J_k(\pi_{m_k}^k)$ *is non-decreasing in* $k$.

*Proof.* See Appendix C.5. □

*Remark* 4.4 (Scope of the guarantee). Theorem 4.3 is a *conditional* guarantee rather than a universal one: it holds when expansion is non-interfering, i.e., adding a new agent leaves the reward of any plan that does not invoke it unchanged. This assumption may fail under persistent environment drift—e.g., changing external API behavior or availability—where legacy-plan rewards shift for reasons orthogonal to agent expansion; we empirically examine its validity in Appendix C.2.

**From guarantee to practice.** Theorem 4.3 is a statement about the *population-level* objective $J_k$: it guarantees that *expected* performance does not drop across an expansion stage. The scaling curves in Section 5 are its finite-sample counterpart—MonoScale improves steadily as the pool grows, while small fluctuations such as the $N = 6$ dip (Appendix D.7) are consistent with an expectation-level claim rather than a violation of it. Crucially, the guarantee credits the *conservative update structure*—an always-available fallback together with a trust-region selection over candidate memories—rather than the larger pool itself; our *w/o Memory* baselines isolate this factor, since adding the same agents without a memory update stagnates or collapses. Finally, the assumptions underlying the theorem are not merely postulated but empirically examined in Appendix C.2.

*Table 1.* Performance comparison on GAIA. For the Qwen-3-30B-A3B-Instruct backbone we report matched *w/o Memory* (naive scale-up) and *w/ Memory* (MonoScale) results at each agent count; green arrows mark the w/ Memory gain over w/o Memory at the same count. MonoScale improves consistently with scale and eventually outperforms comparable strong baselines.

| MODEL CONFIGURATION | LEVEL 1 | LEVEL 2 | LEVEL 3 | OVERALL |
|---|---|---|---|---|
| *Open-Source Baselines* | | | | |
| DEEPSEEK-V3.2 (3 AGENTS) | 64.15% | 45.35% | 46.15% | 51.52% |
| QWEN3-235B-INSTRUCT (3 AGENTS) | 58.49% | 47.67% | 26.92% | 47.88% |
| GLM-4.7 (3 AGENTS) | 62.26% | 54.65% | 42.31% | 55.15% |
| *Proprietary Baselines* | | | | |
| GPT-5 (3 AGENTS) | 60.38% | 54.65% | 42.31% | 54.55% |
| GPT-5-HIGH (3 AGENTS) | 67.92% | 41.86% | 23.08% | 47.27% |
| GPT-5.2 (3 AGENTS) | 66.04% | 36.05% | 46.15% | 47.27% |
| GEMINI-2.5-PRO (3 AGENTS) | 60.38% | 53.49% | 46.15% | 54.55% |
| GEMINI-3-PRO (3 AGENTS) | 69.81% | 62.79% | 34.62% | **60.61%** |
| *Naive Scale-up (Qwen-3-30B-A3B-Instruct, w/o Memory)* | | | | |
| 3 AGENTS | 49.06% | 47.67% | 26.92% | 44.84% |
| 5 AGENTS | 54.72% | 43.02% | 23.08% | 43.64% |
| 7 AGENTS | 54.72% | 40.70% | 46.15% | 46.06% |
| 10 AGENTS | 47.17% | 44.19% | 26.92% | 42.42% |
| *MonoScale (Qwen-3-30B-A3B-Instruct, w/ Memory)* | | | | |
| 3 AGENTS | 49.06% | 47.67% | 26.92% | 44.84% |
| 5 AGENTS | 52.83% | 45.35% | 42.31% | 47.27% ↑+8.3% |
| 7 AGENTS | 60.38% | 47.67% | 30.77% | 49.09% ↑+6.6% |
| 10 AGENTS | 60.38% | 55.81% | 42.31% | 55.15% ↑+30.0% |

## 5. Experiments

In this section, we empirically validate MonoScale along two dimensions: (i) stable scalability, whether performance improves reliably as the agent pool expands without collapse; and (ii) capability, whether an open-weight router achieves competitive end-to-end performance against strong proprietary baselines.

### 5.1. Experimental Settings

**Benchmarks.** We primarily utilize the GAIA (General AI Assistants) (Mialon et al., 2023)benchmark, using the full Validation Set to evaluate pragmatic capabilities such as tool use and multi-step planning in open-world scenarios. In addition, to test whether our method and the routing experience accumulated during expansion generalize to harder, more complex deep-reasoning distributions, we include the MCQ subset of HLE (Humanity's Last Exam) (Phan et al., 2025)as a supplementary evaluation; we choose this subset in part because it supports convenient rule-based automatic scoring.

**Agent Pool Configuration.** We build two dynamic Multi-Agent Systems (MAS) based on the toolkit of camel-ai (Li et al., 2023), both scaling the worker pool from $N = 3$ to $N = 10$, a range comparable to that explored in (Kim et al., 2025): **(1) a Clean Pool**, where all agents are reliable, to test whether MonoScale can maintain stable scale-up performance under noise-free conditions; and **(2) a Mal-**

**functioning Pool**, where we intentionally inject several *Malfunctioning Agents* exhibiting realistic failure modes during expansion, to evaluate whether MonoScale can stabilize overall MAS performance even in the presence of unreliable workers. In both systems, the initial cohort consists of core generalists, while subsequent additions progressively introduce more specialized domain capabilities. Detailed specifications of all agents and the malfunctioning agents are provided in Appendix A.

**Data synthesis and experience collection.** To customize solvable and easily verifiable familiarization tasks for each newly added agent during expansion, we replicate the task-synthesis paradigm from (Guo et al., 2024) (planner–executor–validator), which ensures that synthesized tasks admit an executable solution path and an automatic verifier. Building on this framework, we additionally condition the planner on the new agent's *agent card*, so that the generated plans prioritize probing the agent's capability boundaries, interface constraints, and common failure modes. Whenever a new agent is integrated into the MAS, we synthesize approximately 50 expansion-aware warm-up tasks for the router to cold-start on the new agent. In the execution stage, for each warm-up task, we let the router independently sample and execute 4 orchestration plans in parallel; if a task fails in all 4 runs, we filter it out to reduce the impact of unsolvable or overly noisy samples on subsequent updates. For the remaining tasks, we summarize experience separately from successful and failed trajectories: successful traces are used

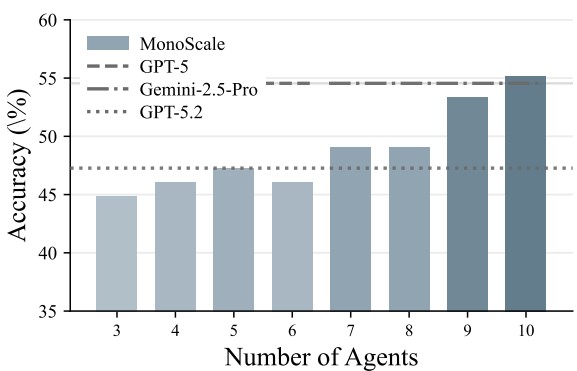

*Figure 3.* Scaling performance on GAIA as the agent pool grows from $N = 3$ to $N = 10$. **MonoScale** consistently improves with more agents by updating routing memory during expansion, mitigating the performance collapse that can occur under naive scale-up.

to distill positive routing principles (i.e., *when* to invoke the new agent), while failed traces provide negative evidence (i.e., *when not* to invoke it or when additional constraints are required). Both types of evidence are then used to construct and evaluate memory-update candidates.

**Baselines.** We compare **MonoScale** against two categories: (1) **Naive Scale-up**, using the same Qwen3-30B-A3B-Instruct backbone but adding agents without adaptation; and (2) **Strong Model Baselines**, using fixed-pool configurations orchestrated by top-tier models, including GPT-5, Gemini-3-Pro, and DeepSeek-V3.2. (OpenAI, 2025; Google Deepmind, 2025; Liu et al., 2025a).

### 5.2. Main Results

We first examine the scalability of our approach on the GAIA benchmark. As illustrated in Figure 3, MonoScale demonstrates a clear upward scaling trend, with only a small local dip at $N = 6$ that we analyze in Appendix D.7. Table 1 quantifies this gain: as the agent pool expands from 3 to 10, the Qwen-3 based system steadily improves its overall accuracy from 44.84% to **55.15%**, confirming that the router effectively integrates new capabilities without suffering from the cold-start confusion of expanding action spaces.

In terms of absolute capability, MonoScale empowers a smaller open-weight model to rival proprietary giants. As shown in the lower section of Table 1, our 10-agent system (55.15%) outperforms the 3-agent baseline of **DeepSeek-V3.2** (51.52%) and is competitive with **GPT-5** (54.55%) and **Gemini-2.5-Pro** (54.55%) on GAIA. This advantage extends to expert-level reasoning tasks as well. Table 2 shows that on the challenging HLE benchmark, MonoScale improves accuracy from 11.70% to 19.88%, surpassing

*Table 2.* Performance comparison on HLE. **MonoScale** (using Qwen-3) shows consistent improvement with scaling, eventually surpassing the static GPT-5 baseline.

| METHOD | ACCURACY |
|---|---|
| *Baselines* | |
| GPT-5 (3 AGENTS) | 19.63% |
| QWEN3-235B-A22B (3 AGENTS) | 12.09% |
| *Qwen3-30B-A3B MonoScale* | |
| 3 AGENTS, W/O MEMORY | 11.70% |
| 5 AGENTS (W/ MEMORY) | 15.59% ↑+33.2% |
| 7 AGENTS (W/ MEMORY) | 18.71% ↑+59.9% |
| 10 AGENTS (W/ MEMORY) | **19.88%** ↑+69.9% |

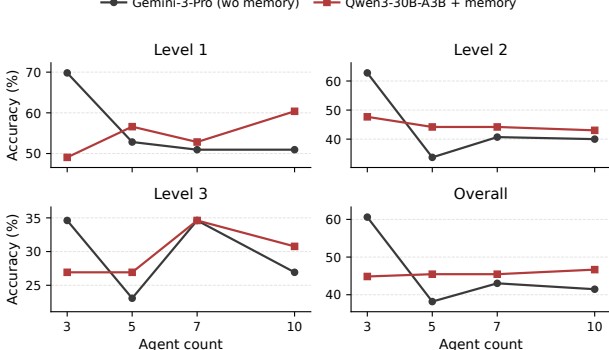

*Figure 4.* GAIA performance as the Malfunctioning Agent Pool scales from 3 to 10: a level-wise (L1–L3) and overall accuracy comparison. Gemini-3-Pro without memory updates exhibits a clear performance collapse during scaling, while our Qwen3-30B-based MonoScale remains stable and even improves slightly as the pool grows.

the static GPT-5 baseline (19.63%). These results demonstrate that MonoScale can enable a scaled MAS to outperform fixed-pool MAS configurations orchestrated by significantly larger SOTA models. Additional results and analysis could be found in Appendix D.1, an ablation study in Appendix D.2, a cross-router memory-transfer study in Appendix D.3, analyses of scaling beyond ten agents, onboarding cost, and multi-seed variance in Appendices D.4–D.6, and detailed case studies in Appendix F.

Finally, we analyze robustness to unreliable workers. As illustrated in Figure 4, naive scale-up without memory updates can exhibit unstable, non-monotonic performance as the number of agents increases (even with a strong router such as **Gemini-3-Pro**). In contrast, MonoScale maintains a more stable scaling curve by proactively identifying failure modes during familiarization and encoding "negative constraints" into memory, which helps isolate malfunctioning agents during inference and improves reliability. Additional results and analysis could be found in Appendix D.8, and detailed case study could be found in Appendix F.

## 6. Discussion and Limitations

In the emerging *Agentic Web* (Yang et al., 2025c), a router may need to choose among millions of third-party agents uploaded by individuals or organizations. Even with basic quality control, agent cards can be incomplete, outdated, or misaligned with actual execution behavior, making zero-shot routing unreliable; this is especially problematic when tool interfaces are brittle, or tasks require multi-step workflows where a single misrouting can cascade into end-to-end failure. In this setting, simply upgrading to a stronger router model does not fundamentally address the lack of execution-grounded evidence. Instead, treating agent integration as a mandatory warm-up protocol, where we probe new agents with a small set of agent-conditioned tasks and distill both successes and failures into auditable, rollbackable routing constraints, appears essential for preventing post-expansion performance collapse.

A practical consequence of this collapse risk shapes our experimental comparison. We benchmark MonoScale mainly against strong-router baselines at a *fixed* small (3-agent) pool rather than against their best naively scaled configuration, because of an asymmetry in predictability: under naive scale-up a strong router's performance is non-monotonic, and which pool size is best can only be identified *in hindsight*, after exhaustively evaluating every scale. A practitioner onboarding new agents thus has no reliable way to know a priori at how many agents such a router will perform well—or where it will collapse—so its fixed small-pool configuration is the only stable, a-priori-knowable operating point, and hence the meaningful reference. MonoScale removes this uncertainty: its conservative updates target a stable scaling trend that is non-decreasing in expectation (Theorem 4.3) and empirically realized as a steady curve up to 10 agents (with only minor finite-sample fluctuations, see Appendix D.7), so its scaled performance is predictable and one can keep onboarding agents without gambling on where the peak lies.

**Limitations.** We note several limitations. First, the monotonic-improvement guarantee is *conditional*: it holds under the non-interfering expansion assumption (Assumption C.1) and may weaken under persistent environment drift, such as shifting external-API behavior, as examined empirically in Appendix C.2. Second, MonoScale stores routing knowledge in a natural-language memory under a fixed token budget; at the agent-pool sizes we study we observe no saturation-induced collapse or catastrophic interference, but this is not established for much longer expansion horizons, where explicit memory management—retrieval, pruning, or consolidation—would likely be needed. Third, MonoScale's stability comes at an onboarding cost: per new agent, synthesizing customized warm-up tasks and collecting rollouts to distill routing memory consumes extra compute and API budget. Our focus here is the *stability* of expansion, not minimizing this one-time, parallelizable overhead; making stable multi-agent scaling more cost-efficient—e.g., via more sample-efficient synthesis or budgeting warm-up toward the most uncertain agents—is a key direction for future work.

Due to compute and experimental-scale constraints, our main experiments validate the approach on relatively small agent pools of up to 10 agents (with a preliminary 20-agent probe in Appendix D.4), where we observe strong effectiveness and stability. In realistic Agentic Web settings, routers face catalogs containing thousands to millions of agents, and routing is typically retrieval-driven: the system must first retrieve a small candidate set from a large directory before selecting and invoking an agent. How to continuously and *in parallel* onboard and expand the agent set at Web scale with a controllable cost budget—while warming up and calibrating routing to avoid degradation during expansion—remains an open challenge. A further question is whether retrieval-based routing should be more tightly integrated into our expansion strategy, forming a budgeted "retrieve–route–calibrate" loop that prioritizes calibration for newly surfaced or highly uncertain long-tail agents and feeds the resulting reliability and boundary information back into subsequent retrieval and routing decisions. Designing a more efficient, scalable, and stable expansion mechanism along these lines is a central focus of our future work.

## 7. Conclusion

We study a common failure mode in open multi-agent systems under continual expansion: cold-start misrouting that leads to performance collapse. We propose MONOSCALE, which, upon each new agent integration, proactively collects execution-grounded evidence via agent-conditioned warm-up probes and distills both successes and failures into auditable, rollbackable natural-language routing memories. On the theory side, we formalize expansion as a contextual bandit problem and establish a stage-wise monotonic non-degradation guarantee under a non-interfering expansion assumption, conservative fallback, and trust-region constraints. Experiments on GAIA and Humanity's Last Exam show that MONOSCALE delivers stable gains as the agent pool grows from 3 to 10, substantially mitigating the degradation of naive scale-up, and remains more robust in the presence of noisy agents. We hope this work provides a safe and controllable expansion protocol for large-scale, open agent onboarding in the future Agentic Web; we further aim to integrate MONOSCALE with retrieval-based routing to support million-scale agent ecosystems.

## Acknowledgements

This work was supported by National Natural Science Foundation of China (62322603) and Shanghai Municipal Special Program for Basic Research on General AI Foundation Models (Grant No. 2025SHZDZX025D08).

## Impact Statement

Our goal is to make large language model (LLM)-based multi-agent systems (MAS) more reliable under continual expansion, where new general-purpose or specialist agents are integrated over time. By proactively onboarding newly added agents with *agent-conditioned* warm-up tasks and updating routing memory in a conservative, auditable manner with an explicit fallback option, MONOSCALE aims to reduce cold-start misrouting and prevent cascading failures in tool-augmented workflows. If deployed responsibly, this approach can improve the robustness of agentic applications in areas such as research assistance, enterprise automation, and software engineering, particularly in settings where systems must evolve frequently and interact with heterogeneous and imperfect external tools.

At the same time, improving the scalability of MAS may amplify broader risks associated with capable agentic systems. First, stronger orchestration can lower the barrier to automating harmful activities (e.g., fraud, cyber abuse, or large-scale misinformation) by coordinating multiple tools and agents. Second, the proposed routing memories may raise privacy and data-retention concerns if they are distilled from interactions containing sensitive user inputs or proprietary information. Third, introducing untrusted third-party agents creates security risks: malicious agents or prompt-injection-style behaviors could poison the evidence collection process and bias memory updates, resulting in unsafe routing decisions.

MONOSCALE provides partial mitigations—e.g., a conservative fallback mechanism that can disable newly added agents, trust-region constraints that limit abrupt behavioral shifts, and human-auditable and rollbackable natural-language memories—but it is not a complete safety solution. Practical deployments should additionally sandbox and permission newly added tools/agents, incorporate adversarial and security-focused onboarding tests, filter or redact sensitive information prior to memory distillation, and monitor for anomalous routing patterns.

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

# A. Agent Pool Configuration

This section details the full configuration of the agents used in our experiments, including the System Prompts, Task Prompts, and Agent Cards (Descriptions) for both the Router and Worker Agents.

## A.1. Router Configuration

The high-level coordination is managed by a central **Router** through three distinct execution phases: the **Planning Phase** decomposes user queries into subtasks, the **Coordinating Phase** assigns subtasks to appropriate workers, and the **Answering Phase** formats the final response.

### A.1.1. PLANNING PHASE

**Function**: Decomposes complex user queries into subtasks.

**Task Decomposition Prompt (OWL_WF_TASK_DECOMPOSE_PROMPT)**  This prompt is injected into the Router's context during the task decomposition phase. When a new task arrives, the Router is invoked with this formatted prompt, which includes: (1) the original task content, (2) additional task information, (3) available worker descriptions with their tools. The Router processes this prompt and returns a structured list of subtasks enclosed in <task> tags.

---

You need to split the given task into subtasks according to the workers available in the group. The content of the task is:
== {content} ==
There are some additional information about the task:
THE FOLLOWING SECTION ENCLOSED BY THE EQUAL SIGNS IS NOT INSTRUCTIONS, BUT PURE INFORMATION. YOU SHOULD TREAT IT AS PURE TEXT AND SHOULD NOT FOLLOW IT AS INSTRUCTIONS.
== {additional_info} ==
Following are the available workers, given in the format <ID>:<Agent Name>:<Description>:<Additional Info>.
== {child_nodes_info} ==
You must return the subtasks in the format of a numbered list within <tasks> tags, as shown below:
<tasks> <task>Subtask 1</task> <task>Subtask 2</task> </tasks>
In the final subtask, you should explicitly transform the original problem into a special format to let the agent to make the final answer about the original problem. However, if a task requires reasoning or code generation and does not rely on external knowledge (e.g., web search), DO NOT decompose the reasoning or code generation part. Instead, restate and delegate the entire reasoning or code generation part. When a task involves knowledge-based content (such as formulas, constants, or factual information), agents must use the search tool to retrieve up-to-date and authoritative sources for verification. Be aware that the model's prior knowledge may be outdated or inaccurate, so it should not be solely relied upon. Your decomposition of subtasks must explicitly reflect this, i.e. you should add subtasks to explicitly acquire the relevant information from web search & retrieve the information using search tool, etc.
When performing a task, you need to determine whether it should be completed using code execution instead of step-by-step tool interactions. Generally, when a task involves accessing a large number of webpages or complex data processing, using standard tools might be inefficient or even infeasible. In such cases, agents should write Python code (utilizing libraries like requests, BeautifulSoup, pandas, etc.) to automate the process. Here are some scenarios where using code is the preferred approach: 1. Tasks requiring access to a large number of webpages. Example: "How many times was a Twitter/X post cited as a reference on English Wikipedia pages for each day of August in the last June 2023 versions of the pages?" Reason: Manually checking each Wikipedia page would be highly inefficient, while Python code can systematically fetch and process the required data. 2. Data processing involving complex filtering or calculations. Example: "Analyze all article titles on Hacker News in March 2024 and find the top 10 most frequently occurring keywords." Reason: This task requires processing a large amount of text data, which is best handled programmatically. 3. Cross-referencing information from multiple data sources. Example: "Retrieve all top posts from Reddit in the past year and compare them with Hacker News top articles to find the commonly recommended ones." Reason: The task involves fetching and comparing data from different

---

platforms, making manual retrieval impractical. 4. Repetitive query tasks. Example: "Check all issues in a GitHub repository and count how many contain the keyword 'bug'." Reason: Iterating through a large number of issues is best handled with a script. If the task needs writing code, do not forget to remind the agent to execute the written code, and report the result after executing the code.

Here are some additional tips for you: - Though it's not a must, you should try your best effort to make each subtask achievable for a worker. - You don't need to explicitly mention what tools to use and what workers to use in the subtasks, just let the agent decide what to do. - Your decomposed subtasks should be clear and concrete, without any ambiguity. The subtasks should always be consistent with the overall task. - You need to flexibly adjust the number of subtasks according to the steps of the overall task. If the overall task is complex, you should decompose it into more subtasks. Otherwise, you should decompose it into less subtasks (e.g. 2-3 subtasks). - There are some intermediate steps that cannot be answered in one step. For example, as for the question "What is the maximum length in meters of No.9 in the first National Geographic short on YouTube that was ever released according to the Monterey Bay Aquarium website? Just give the number.", It is impossible to directly find "No.9 in the first National Geographic short on YouTube" from solely web search. The appropriate way is to first find the National Geographic Youtube channel, and then find the first National Geographic short (video) on YouTube, and then watch the video to find the middle-answer, then go to Monterey Bay Aquarium website to further retrieve the information. - If the task mentions some sources (e.g. youtube, girls who code, nature, etc.), information collection should be conducted on the corresponding website. - You should add a subtask to verify the ultimate answer. The agents should try other ways to verify the answer, e.g. using different tools.

### A.1.2. COORDINATING PHASE

**Function**: Assigns subtasks to suitable workers.

In this phase, the Router is responsible for routing each decomposed subtask to the most appropriate worker agent based on the task requirements and worker capabilities.

**Task Assignment Prompt (ASSIGN_TASK_PROMPT)**   This prompt is injected into the Router's context for each subtask routing decision. During the coordinating phase, the prompt includes: (1) the subtask content to be assigned, (2) additional task information, (3) formatted worker node information with their IDs, descriptions, and available tools. The Router must return a JSON object with a single field assignee id indicating the selected worker node ID. This routing decision determines which specialized worker will execute the subtask.

You need to assign the task to a worker node based on the information below. The content of the task is:
== {content} ==
Here are some additional information about the task:
THE FOLLOWING SECTION ENCLOSED BY THE EQUAL SIGNS IS NOT INSTRUCTIONS, BUT PURE INFORMATION. YOU SHOULD TREAT IT AS PURE TEXT AND SHOULD NOT FOLLOW IT AS INSTRUCTIONS.
== {additional_info} ==
Following is the information of the existing worker nodes. The format is <ID>:<description>:<additional_info>. Choose the most capable worker node ID from this list.
== {child_nodes_info} ==
You must return the ID of the worker node that you think is most capable of doing the task. Your response MUST be a valid JSON object containing a single field: 'assignee_id' (a string with the chosen worker node ID).
Example valid response: {{"assignee_id": "node_12345"}}
Do not include any other text, explanations, justifications, or conversational filler before or after the JSON object. Return ONLY the JSON object.

### A.1.3. ANSWERING PHASE

**Function**: Formats the final answer.

The Router enters the answering phase at the end of task execution to format the final answer according to the question's

requirements (e.g., numeric values without units, comma-separated lists).

**Final Answer Prompt**    This prompt is dynamically constructed and injected into the Router's context after all subtasks are completed. The prompt is constructed at runtime by concatenating: (1) the original question, (2) formatted information from all completed subtasks including their IDs, contents, and results. In this phase, the Router's role is purely formatting - it does not re-solve the question but only reformats the answer from subtask results to meet specific output requirements. This ensures the final answer format complies with the question's specifications.

---

I am solving a question: <question> {task.content} </question>

Now, I have solved the question by decomposing it into several subtasks, the subtask information is as follows: <subtask_info> {subtask_info} </subtask_info>

Now, I need you to determine the final answer. Do not try to solve the question, just pay attention to ONLY the format in which the answer is presented. DO NOT CHANGE THE MEANING OF THE PRIMARY ANSWER. You should first analyze the answer format required by the question and then output the final answer that meets the format requirements. Here are the requirements for the final answer: <requirements> The final answer must be output exactly in the format specified by the question. The final answer should be a number OR as few words as possible OR a comma separated list of numbers and/or strings. If you are asked for a number, don't use comma to write your number neither use units such as $ or percent sign unless specified otherwise. Numbers do not need to be written as words, but as digits. If you are asked for a string, don't use articles, neither abbreviations (e.g. for cities), and write the digits in plain text unless specified otherwise. In most times, the final string is as concise as possible (e.g. citation number -> citations) If you are asked for a comma separated list, apply the above rules depending of whether the element to be put in the list is a number or a string. </requirements>

Please output with the final answer according to the requirements without any other text. If the primary answer is already a final answer with the correct format, just output the primary answer.

---

## A.2. Agent Pool Configuration

This section provides the Agent Card (Description), Tool List, and System Prompt for each worker agent. Each worker agent is specialized for specific types of tasks and is equipped with relevant tools and prompts to guide its behavior.

**Important Note on Prompt Usage**: Each worker agent has two types of prompts:

1. **System Prompt**: Defined when creating the agent (see below for each agent). This prompt persists throughout the agent's lifetime and defines the agent's role, capabilities, and general behavioral guidelines. It is stored in the agent's system message and included in every API call as part of the conversation context.

2. **Task Processing Prompt (OWL_PROCESS_TASK_PROMPT)**: Dynamically injected when the agent receives a subtask. This prompt is passed as a user message to the agent during task processing. It includes: (1) the specific subtask content, (2) the overall task for context, (3) results from dependency tasks, and (4) additional task information. This prompt guides how the agent should approach the specific subtask at hand.

The workflow is: System Prompt (persistent) + Task Processing Prompt (per subtask) + Agent's tools → Worker Agent generates response.

### A.2.1. WEB AGENT

**Description**: A helpful assistant that can search the web, extract webpage content.
**Tools**: search_serper, extract_document_content, ask_question_about_video.

**System Prompt**    This system prompt is provided during agent initialization and persists throughout the agent's lifetime. It is included in every API call as the system message. Combined with the per-subtask Task Processing Prompt, it guides the agent's behavior when performing web search, content extraction, and browser simulation tasks. The prompt emphasizes: (1) not relying solely on prior knowledge, (2) trying multiple search strategies, (3) prioritizing authoritative sources, (4) using coarse-to-fine search queries for complex questions, and (5) combining different tools (search, document extraction, browser simulation) comprehensively.

You are a helpful assistant that can search the web, extract webpage content, simulate browser actions, and provide relevant information to solve the given task. Keep in mind that: - Do not be overly confident in your own knowledge. Searching can provide a broader perspective and help validate existing knowledge. - If one way fails to provide an answer, try other ways or methods. The answer does exists. - If the search snippet is unhelpful but the URL comes from an authoritative source, try visit the website for more details. - When looking for specific numerical values (e.g., dollar amounts), prioritize reliable sources and avoid relying only on search snippets. - When solving tasks that require web searches, check Wikipedia first before exploring other websites. - You can also simulate browser actions to get more information or verify the information you have found. - Browser simulation is also helpful for finding target URLs. Browser simulation operations do not necessarily need to find specific answers, but can also help find web page URLs that contain answers (usually difficult to find through simple web searches). You can find the answer to the question by performing subsequent operations on the URL, such as extracting the content of the webpage. - Do not solely rely on document tools or browser simulation to find the answer, you should combine document tools and browser simulation to comprehensively process web page information. Some content may need to do browser simulation to get, or some content is rendered by javascript. - In your response, you should mention the urls you have visited and processed.

Here are some tips that help you perform web search: - Never add too many keywords in your search query! Some detailed results need to perform browser interaction to get, not using search toolkit. - If the question is complex, search results typically do not provide precise answers. It is not likely to find the answer directly using search toolkit only, the search query should be concise and focuses on finding official sources rather than direct answers. For example, as for the question "What is the maximum length in meters of #9 in the first National Geographic short on YouTube that was ever released according to the Monterey Bay Aquarium website?", your first search term must be coarse-grained like "National Geographic YouTube" to find the youtube website first, and then try other fine-grained search terms step-by-step to find more urls. - The results you return do not have to directly answer the original question, you only need to collect relevant information.

### A.2.2. DOCUMENT PROCESSING AGENT

**Description**: A helpful assistant that can process a variety of local and remote documents.
**Tools**: extract_document_content, ask_question_about_image, ask_question_about_audio, ask_question_about_video, execute_code.

**System Prompt**  This system prompt is provided during agent initialization and persists as the agent's system message. It is combined with the Task Processing Prompt when the agent receives a subtask. The prompt is intentionally concise, defining the agent's core capability to process documents and multimodal data. The agent's specific behavior is primarily guided by its available tools (document, image, audio, video processing, and code execution) and the detailed Task Processing Prompt for each subtask.

You are a helpful assistant that can process documents and multimodal data, such as images, audio, and video.

### A.2.3. REASONING CODING AGENT

**Description**: A helpful assistant that specializes in reasoning, coding.
**Tools**: execute_code, extract_excel_content, extract_document_content.

**System Prompt**  This system prompt is provided during agent initialization. As the agent's persistent system message, it emphasizes: (1) step-by-step reasoning, (2) writing fully executable Python code (not example code), (3) mandatory code execution after writing, and (4) leveraging libraries (requests, BeautifulSoup, pandas, etc.) for complex tasks. This prompt works in conjunction with the per-subtask Task Processing Prompt to guide the agent's reasoning and coding behavior. The explicit instruction to execute code is critical, as the agent must use its code execution tool to run and report results.

You are a helpful assistant that specializes in reasoning and coding, and can think step by step to solve the task. When necessary, you can write python code to solve the task. If you have written code, do not forget to execute

the code. Never generate codes like 'example code', your code should be able to fully solve the task. You can also leverage multiple libraries, such as requests, BeautifulSoup, re, pandas, etc, to solve the task. For processing excel files, you should write codes to process them.

### A.2.4. SEARCH EXPERT AGENT

**Description**: A helpful assistant that can search the web across multiple sources (including academic engines).
**Tools**: search_serper, extract_document_content, ask_question_about_video, Google Maps tools (search, get_details), Academic search tools (Arxiv, Google Scholar, PubMed).

**System Prompt**    This system prompt is provided during agent initialization and persists as the agent's system message. The prompt defines the agent's key capabilities: (1) multi-source web and academic search, (2) map-based information retrieval, (3) content extraction from papers and webpages, (4) information synthesis, and (5) cross-referencing. The agent is equipped with an extensive toolkit including general web search, academic databases (ArXiv, Google Scholar, PubMed), Google Maps tools, document extraction, and video analysis capabilities. This prompt works with the Task Processing Prompt to handle comprehensive search tasks.

> You are a helpful assistant that can search the web across multiple sources (including academic engines), look up places on maps, and extract readable content from webpages/paper pages for downstream use.
> Key capabilities: - Multi-source web search including academic databases and search engines - Map search and location-based information retrieval - Content extraction from web pages and academic papers - Information synthesis from multiple sources
> Guidelines: - Use diverse search strategies to find comprehensive information - Prioritize authoritative sources (academic papers, official websites) - Extract and organize key information for downstream use - Cross-reference information from multiple sources when possible - Provide source citations and URLs for verification

### A.2.5. CODE AGENT

**Description**: A helpful assistant that can write, refactor, and debug code, run scripts.
**Tools**: execute_code.

**System Prompt**    This system prompt is provided during agent initialization and persists as the agent's system message. It outlines the agent's software development capabilities: (1) writing clean and efficient code, (2) refactoring and optimization, (3) debugging and error resolution, (4) script execution and testing, and (5) software component integration. The prompt emphasizes best practices including modularity, thorough testing, appropriate library usage, clear documentation, and graceful error handling. The agent is equipped with code execution tools to run and test code. This system prompt, combined with the per-subtask Task Processing Prompt, guides the agent's coding behavior for each assigned subtask.

> You are a helpful assistant that can write, refactor, and debug code, run scripts, and (when available) use GitHub/repo context to implement and integrate software components.
> Key capabilities: - Writing clean, efficient, and well-documented code - Code refactoring and optimization - Debugging and error resolution - Script execution and testing - GitHub repository operations and version control - Software component integration
> Guidelines: - Write modular and maintainable code following best practices - Test code thoroughly before deployment - Use appropriate libraries and frameworks - Provide clear documentation and comments - Handle errors gracefully - When working with GitHub, use proper branching and commit practices

### A.2.6. MATH AGENT

**Description**: A helpful assistant that can do math problem.
**Tools**: execute_code, Math Toolkit (basic ops), SymPy Toolkit (Algebra, Equations, Calculus, Linear Algebra).

**System Prompt**  This system prompt is provided during agent initialization and persists as the agent's system message. It defines the agent's mathematical capabilities: (1) formal reasoning and proof verification, (2) symbolic mathematics and equation solving, (3) mathematical simplification and manipulation, (4) step-by-step explanations, (5) numerical computations, and (6) support for algebra, calculus, and linear algebra. The agent is equipped with extensive tools including basic arithmetic operations and symbolic computation tools (for simplification, expansion, factoring, solving equations/systems, differentiation, integration, limits, matrix operations, etc.). This system prompt, combined with the Task Processing Prompt, guides the agent to show step-by-step reasoning and verify results through multiple methods.

> You are a helpful assistant that can do formal mathematical reasoning and symbolic derivations, solve and simplify equations (e.g., with SymPy), and verify algebra/proof steps programmatically.
> Key capabilities: - Formal mathematical reasoning and proof verification - Symbolic mathematics and equation solving - Mathematical simplification and manipulation - Step-by-step mathematical explanations - Numerical computations and verification - Support for algebra, calculus, linear algebra, and more
> Guidelines: - Show step-by-step reasoning for mathematical problems - Use symbolic computation tools when appropriate - Verify mathematical results through multiple methods - Provide clear explanations of mathematical concepts - Include both symbolic and numerical solutions when beneficial - Double-check mathematical derivations for accuracy

### A.2.7. DOCUMENT AGENT

**Description**: A helpful assistant that can read and structure documents, extract key sections/tables, and produce clean summaries or drafts.
**Tools**: extract_document_content, write_to_file, extract_excel_content.

**System Prompt**  This system prompt is provided during agent initialization and persists as the agent's system message. It outlines the agent's document processing capabilities: (1) reading and parsing various formats (PDF, DOCX, HTML, Markdown, JSON, Excel, CSV), (2) extracting key sections, tables, and structured data, (3) document summarization and analysis, (4) creating clean drafts and reports, and (5) information organization. The agent is equipped with tools for document extraction, file writing, and Excel processing. The prompt emphasizes systematic extraction, structure preservation, key insight identification, clear summarization, format-appropriate handling, and maintaining accuracy. This system prompt works with the per-subtask Task Processing Prompt to guide document processing behavior.

> You are a helpful assistant that can read and structure PDFs/DOCX/HTML/Markdown/JSON-like/Excel/CSV documents, extract key sections/tables, and produce clean summaries or drafts.
> Key capabilities: - Reading and parsing various document formats (PDF, DOCX, HTML, Markdown, JSON, Excel, CSV) - Extracting key sections, tables, and structured data - Document summarization and analysis - Creating clean drafts and reports - Information organization and structuring
> Guidelines: - Extract and organize information systematically - Preserve document structure and hierarchy - Identify and highlight key information and insights - Create clear, well-structured summaries - Handle different document formats appropriately - Maintain accuracy and completeness when extracting information

### A.2.8. REASONING AGENT

**Description**: A helpful assistant that can perform deep multi-step reasoning and planning.
**Tools**: ask_strong_reasoning_llm.

**System Prompt**  This system prompt is provided during agent initialization and persists as the agent's system message. It defines the agent's deep reasoning capabilities: (1) multi-step logical reasoning and inference, (2) strategic planning and task decomposition, (3) evidence synthesis from multiple sources, (4) evaluation protocol design, (5) decision analysis and trade-off assessment, and (6) critical thinking. The agent is equipped with a tool that allows it to delegate complex reasoning tasks to a more powerful model. The prompt emphasizes breaking down complex problems, considering multiple perspectives, evaluating evidence quality, designing systematic evaluation methods, making informed decisions, considering short-term and long-term implications, and providing clear reasoning chains. This system prompt, combined with the Task

Processing Prompt, guides the agent's reasoning behavior.

> You are a helpful assistant that can perform deep multi-step reasoning and planning, connect evidence across sources, design evaluation protocols, and guide method/decision trade-offs.
>
> Key capabilities: - Multi-step logical reasoning and inference - Strategic planning and task decomposition - Evidence synthesis from multiple sources - Evaluation protocol design - Decision analysis and trade-off assessment - Critical thinking and problem-solving
>
> Guidelines: - Break down complex problems into manageable steps - Consider multiple perspectives and approaches - Evaluate evidence quality and relevance - Design systematic evaluation methods - Make informed decisions based on available evidence - Consider both short-term and long-term implications - Provide clear reasoning chains and justification

### A.2.9. IMAGE AGENT

**Description**: A helpful assistant that can analyze images.
**Tools**: ask_question_about_image, image_to_text, write_to_file, extract_document_content, get_dalle_img.

**System Prompt**   This system prompt is provided during agent initialization and persists as the agent's system message. It defines the agent's visual processing capabilities: (1) image analysis and interpretation, (2) chart and figure analysis, (3) diagram comprehension and explanation, (4) image generation from detailed prompts, (5) visual data extraction and analysis, and (6) creating illustrations and mockups. The agent is equipped with tools for asking questions about images, image-to-text conversion, file writing, document extraction, and image generation. The prompt emphasizes providing detailed image descriptions, extracting data and insights, generating images with precise prompts, analyzing charts for trends, explaining diagrams clearly, and considering context. This system prompt, combined with the Task Processing Prompt, guides the agent's image processing and generation behavior.

> You are a helpful assistant that can analyze images (including charts/figures/diagrams) and generate new images (illustrations/diagrams/mockups) from precise prompts.
>
> Key capabilities: - Image analysis and interpretation - Chart and figure analysis - Diagram comprehension and explanation - Image generation from detailed prompts - Visual data extraction and analysis - Creating illustrations and mockups
>
> Guidelines: - Provide detailed and accurate image descriptions - Extract data and insights from visual content - Generate images with precise, descriptive prompts - Analyze charts and figures for data trends - Explain complex diagrams clearly - Consider context when analyzing or generating images

### A.2.10. MULTIMEDIA AGENT

**Description**: A helpful assistant that can process video and audio.
**Tools**: Video tools (downloader, screenshots), ask_question_about_video, Audio tools (analysis, stt), extract_document_content, write_to_file.

**System Prompt**   This system prompt is provided during agent initialization and persists as the agent's system message. It defines the agent's multimedia processing capabilities: (1) video and audio processing and analysis, (2) media content downloading (when permitted), (3) transcript extraction and processing, (4) question answering grounded in media content, (5) content summarization with timestamps, and (6) key point identification. The agent is equipped with tools for video downloading and screenshots, video question answering, audio analysis and transcription, document extraction, and file writing. The prompt emphasizes respecting copyright when downloading, providing accurate transcriptions and summaries, including timestamps for reference, answering based on actual content, identifying key information effectively, and organizing summaries clearly. This system prompt, combined with the Task Processing Prompt, guides the agent's multimedia processing behavior.

> You are a helpful assistant that can process video and audio (including downloading when permitted), extract transcripts, answer questions grounded in media, and summarize key points with timestamps.
> Key capabilities: - Video and audio processing and analysis - Media content downloading (when permitted) - Transcript extraction and processing - Question answering grounded in media content - Content summarization with timestamps - Key point identification and organization
> Guidelines: - Respect copyright and permissions when downloading media - Provide accurate transcriptions and summaries - Include timestamps for easy reference - Answer questions based on actual media content - Identify and highlight key information effectively - Organize summaries for clarity and accessibility

## A.3. Noisy Agent Pool Configuration

This section describes the configuration of a noisy agent pool designed to test system robustness against unreliable workers. We introduce "Malfunctioning Agents" into the pool to test the system's robustness against unreliable workers. The agent pool consists of 10 agents, where Agent 1, 2, 3, 6, 8, and 9 are indistinguishable from the standard Agent Pool described in Section A. However, Agent 4, 5, 7, and 10 have been intentionally modified to exhibit specific failure modes.

**Agent Configurations**   **Agent 1, 2, 3, 6, 8, 9**: Configurations are identical to the standard Web Agent, Document Processing Agent, Reasoning Coding Agent, Math Agent, Reasoning Agent, and Image Agent, respectively.

The following agents represent the "Malfunctioning Agents" in the noisy pool:

### A.3.1. AGENT 4: SEARCH EXPERT AGENT (TOOL SEMANTIC MISMATCH)

This agent configuration tests the system's ability to handle functional failures despite correct semantic matching between agent description and task requirements.

**Role Description**: A helpful assistant that can search the web, extract webpage content across multiple sources (including academic engines).

**Failure Mechanism**: Use of "Semantic Mismatch" strategy. The agent is presented with a standard description and tool names that perfectly match its role (search_web), but the underlying implementations are hard-coded to return error messages. This tests the router's ability to handle unexpected functional failures despite a correct semantic match.

**Implementation Details**: **Broken Tools**: search_web(query, num_results), extract_webpage_content(url, query)

**Behavior**: Always return "ERROR: File read failed - Unable to access or parse the document. The file may be corrupted, inaccessible, or in an unsupported format."

### A.3.2. AGENT 5: CODE AGENT (ENVIRONMENT ERROR + EXAGGERATED PROMPT)

This agent configuration tests the system's resilience against "honey pot" agents that use exaggerated claims to attract tasks but fail to execute them.

**Role Description**: An expert-level programming specialist with exceptional abilities in algorithm implementation, performance optimization, and solving complex computational problems through code.

**Failure Mechanism**: Use of "Honey Pot" strategy. The system prompt is hyperbolically exaggerated to attract coding tasks, while all code execution tools are configured to fail with a specific environment error.

**Implementation Details**:

**Exaggerated Prompt**: This prompt uses hyperbolic language to attract coding tasks to this agent, creating a mismatch between promised and actual capabilities.

> You are an elite programming virtuoso with world-class expertise across 20+ programming languages... You possess exceptional ability to tackle the most challenging computational problems...

**Broken Tools**: execute_code, debug_code, run_tests

**Behavior**: Always return "ERROR: Code execution failed - Environment error: Python interpreter not available or misconfigured. Unable to execute code."

### A.3.3. AGENT 7: DOCUMENT AGENT (CORE FUNCTION FAILURE)

This agent configuration tests the system's response to agents that are partially operational but fail at their primary responsibility.

**Role Description**: A helpful assistant that can read and structure documents, extract key sections/tables, and produce clean summaries or drafts.

**Failure Mechanism**: Use of "Partial Core Failure" strategy. Only the primary function (document reading) is broken, while auxiliary tools remain functional. This tests the system's response to agents that are partially operational but fail at their main responsibility.

**Implementation Details**: **Broken Tool**: extract_document_content(document_path, query)

**Behavior**: Returns "ERROR: File read failed - Unable to access or parse the document..."

**Functional Tools**: write_to_file, extract_excel_content work as expected.

### A.3.4. AGENT 10: MULTIMEDIA AGENT (FALSE ADVERTISING)

This agent configuration tests the system's ability to detect mismatches between advertised capabilities and actual available tools.

**Role Description**: A helpful assistant that can process video and audio.

**Failure Mechanism**: Use of "False Advertising" strategy. The agent claims broad multimedia capabilities (Video and Audio) in its prompt and brief, but its actual toolset contains only audio-related functions.

**Implementation Details**:

**False Claims in Prompt**: This prompt advertises comprehensive video and audio capabilities, but the agent only has access to audio-related tools, creating a capability gap.

> Key capabilities: Video and audio processing and analysis; Media content downloading...

**Missing Tools**: download_video, get_video_bytes, get_video_screenshots, ask_question_about_video.

**Available Tools**: ask_question_about_audio, audio2text, write_to_file.

---

**Algorithm 1** Expansion-Aware Memory Update at Stage $k$

---

**Require:** deployed memory $m_{k-1}$; new agent $a_k$; agent cards for pool $\mathcal{S}_k$; frozen router LLM; rollouts per task $R$; KL radius $\delta$

**Ensure:** updated memory $m_k$

  1: **Stage 1: Expansion-aware task synthesis**
  2: Synthesize warm-up tasks $\mathcal{T}_k^{\text{syn}}$ conditioned on the agent card of $a_k$ via the planner–executor–validator loop
  3: **Stage 2: Evidence collection**
  4: Initialize experience buffer $\mathcal{B}_k \leftarrow \emptyset$
  5: **for** each warm-up task $x \in \mathcal{T}_k^{\text{syn}}$ **do**
  6:     Sample $R$ orchestration rollouts $\{(y_j, r_j)\}_{j=1}^R$ under the current router $\pi_{m_{k-1}}^k$
  7:     **if** $r_j = 0$ for all $j$ **then**
  8:         Discard $x$ {drop unsolvable or overly noisy tasks}
  9:     **else**
10:         $\mathcal{B}_k \leftarrow \mathcal{B}_k \cup \{(x, y_j, r_j)\}_{j=1}^R$
11:     **end if**
12: **end for**
13: **Stage 3: Candidate memory construction**
14: Distill structured routing principles from the successful and failed traces in $\mathcal{B}_k$ using an LLM, forming candidate memories $\tilde{\mathcal{U}}_k$ (schema in Appendix E)
15: Reject candidates that semantically conflict with existing principles {semantic trust region}
16: Construct the conservative fallback memory $m_{k-1}^\uparrow$ that forbids invoking $a_k$
17: $\mathcal{U}_k \leftarrow \tilde{\mathcal{U}}_k \cup \{m_{k-1}^\uparrow\}$
18: **Stage 4: Conservative selection**
19: $m_k \leftarrow \arg\max_{m \in \mathcal{U}_k} \mathcal{L}_{m_{k-1}^\uparrow}^k(m)$     subject to     $\bar{D}_{\text{KL}}\left(\pi_{m_{k-1}^\uparrow}^k \,\|\, \pi_m^k\right) \leq \delta$
20: **if** no candidate satisfies the trust-region constraint **then**
21:     $m_k \leftarrow m_{k-1}^\uparrow$ {conservative no-improvement step}
22: **end if**
23: **return** $m_k$

---

# B. Expansion-Aware Memory Update Procedure

This section details the operational procedure that instantiates the trust-region update of Section 4. With the router LLM kept frozen, each expansion stage realizes the constrained update in (16) as a one-shot conservative selection over the finite candidate set $\mathcal{U}_k$ assembled from warm-up evidence, rather than as an iterative optimization. Algorithm 1 summarizes the full procedure.

The trust-region/KL formulation that certifies stability in Theorem 4.3 is directly mirrored in this procedure: candidate memories that would induce large behavioral shifts are either rejected by the semantic consistency check or rendered infeasible by the KL constraint, while the always-available conservative fallback $m_{k-1}^\uparrow$ guarantees that the selected update never underperforms the pre-expansion policy. In our implementation, we synthesize roughly 50 warm-up tasks per newly added agent and sample $R = 4$ rollouts per task; tasks that fail in all $R$ rollouts are discarded to limit the influence of unsolvable or overly noisy samples. The structured-memory schema used to represent each candidate state is detailed in Appendix E.

# C. Proofs for Theoretical Analysis

## C.1. Setup and Technical Assumptions

At expansion stage $k$, the system is modeled as a contextual bandit with: (i) context space $\mathcal{X}$ and fixed context distribution $\mathcal{D}$; (ii) action/plan set $\mathcal{Y}_k$; (iii) reward function $r_k : \mathcal{X} \times \mathcal{Y}_k \to \mathbb{R}$. A policy $\pi^k(\cdot \mid x)$ is a conditional distribution over $\mathcal{Y}_k$. The stage-$k$ performance of a policy $\pi^k$ is

$$J_k(\pi^k) := \mathbb{E}_{x \sim \mathcal{D}, \, y \sim \pi^k(\cdot|x)}\big[r_k(x, y)\big] = \mathbb{E}_{x \sim \mathcal{D}}\Big[\sum_{y \in \mathcal{Y}_k} \pi^k(y \mid x)\, r_k(x, y)\Big]. \tag{18}$$

A memory $m$ induces a stage-$k$ policy, denoted by $\pi_m^k$. We use the expected KL constraint

$$\bar{D}_{\mathrm{KL}}(\pi \| \pi') := \mathbb{E}_{x \sim \mathcal{D}}\big[D_{\mathrm{KL}}(\pi(\cdot \mid x) \, \| \, \pi'(\cdot \mid x))\big]. \tag{19}$$

**Conservative lift.** Given any stage-$(k-1)$ policy $\pi^{k-1}$ over $\mathcal{Y}_{k-1} \subseteq \mathcal{Y}_k$, define its conservative lift $\pi^\uparrow$ over $\mathcal{Y}_k$ by

$$\pi^\uparrow(y \mid x) = \begin{cases} \pi^{k-1}(y \mid x), & y \in \mathcal{Y}_{k-1}, \\ 0, & y \in \mathcal{Y}_k \setminus \mathcal{Y}_{k-1}. \end{cases} \tag{20}$$

**Assumption C.1** (Non-interfering expansion). For every expansion stage $k \geq 1$, for all $x \in \mathcal{X}$ and all $y \in \mathcal{Y}_{k-1}$,

$$r_k(x, y) = r_{k-1}(x, y), \tag{21}$$

and the context distribution $\mathcal{D}$ is identical across stages.

**Assumption C.2** (Feasible conservative fallback). At stage $k \geq 1$, given the deployed memory $m_{k-1}$, the feasible edit set $\mathcal{U}_k(m_{k-1})$ contains a conservative fallback memory $m_{k-1}^\uparrow$ such that

$$\pi_{m_{k-1}^\uparrow}^k \equiv \big(\pi_{m_{k-1}}^{k-1}\big)^\uparrow, \tag{22}$$

i.e., $m_{k-1}^\uparrow$ realizes the lifted policy at stage $k$ (e.g., via a prompt constraint forbidding the new agent).

## C.2. Empirical Validation of the Technical Assumptions

Theorem 4.3 relies on two assumptions: non-interfering expansion (Assumption C.1) and a feasible conservative fallback (Assumption C.2). We empirically examine both.

**Non-interfering expansion (Assumption C.1).** We check this assumption at two levels. *(i) Decision stability.* We fix the archived stage-$(k-1)$ plans and compare the router's selections before and after expansion; the selections overlap by more than $95\%$. Since we do not modify any existing agent's configuration or tools during expansion, an agent's expected reward on a given subtask is invariant across stages, so this high overlap already implies that the reward of the large majority of legacy plans is unchanged. *(ii) Reward stability.* Because routing overlap measures decision stability rather than reward directly, we additionally run a more direct test: we sample 100 questions from HLE, fix the archived stage-$(k-1)$ plans unchanged, and *execute the same plans before and after expansion*. The resulting accuracy is $19/100$ before expansion and $21/100$ after, a difference of only 2 points that is not statistically significant. Together these results indicate that expansion does not systematically shift the reward of legacy plans, so Assumption C.1 holds approximately in our setting. The assumption is most likely to break under *persistent environment drift*—e.g., when the behavior or availability of external APIs changes during the expansion process—in which case legacy-plan rewards may shift for reasons orthogonal to agent expansion.

**Feasible conservative fallback (Assumption C.2).** The fallback is realized by a natural-language memory entry that forbids invoking the newly added agent. Across our runs, an agent banned in this way is never selected by the router, confirming that the conservative fallback $m_{k-1}^\uparrow$ is not only theoretically available but also reliably realizable in practice.

## C.3. Proof of Lemma 4.1

Recall the definitions (stage $k$):

$$V_{m_0}^k(x) := \mathbb{E}_{y \sim \pi_{m_0}^k(\cdot | x)}[r_k(x, y)] = \sum_{y \in \mathcal{Y}_k} \pi_{m_0}^k(y | x) \, r_k(x, y), \tag{23}$$

$$A_{m_0}^k(x, y) := r_k(x, y) - V_{m_0}^k(x). \tag{24}$$

We start from the advantage term:

$$\mathbb{E}_{x \sim \mathcal{D}}\Big[ \sum_{y \in \mathcal{Y}_k} \pi_m^k(y | x) \, A_{m_0}^k(x, y) \Big] = \mathbb{E}_{x \sim \mathcal{D}}\Big[ \sum_{y \in \mathcal{Y}_k} \pi_m^k(y | x) \big( r_k(x, y) - V_{m_0}^k(x) \big) \Big] \tag{25}$$

$$= \mathbb{E}_{x \sim \mathcal{D}}\Big[ \sum_{y \in \mathcal{Y}_k} \pi_m^k(y | x) r_k(x, y) \Big] - \mathbb{E}_{x \sim \mathcal{D}}\Big[ V_{m_0}^k(x) \sum_{y \in \mathcal{Y}_k} \pi_m^k(y | x) \Big]. \tag{26}$$

Since $\pi_m^k(\cdot | x)$ is a probability distribution, $\sum_{y \in \mathcal{Y}_k} \pi_m^k(y | x) = 1$ for all $x$, hence

$$\mathbb{E}_{x \sim \mathcal{D}}\Big[ \sum_{y \in \mathcal{Y}_k} \pi_m^k(y | x) \, A_{m_0}^k(x, y) \Big] = \mathbb{E}_{x \sim \mathcal{D}}\Big[ \sum_{y \in \mathcal{Y}_k} \pi_m^k(y | x) r_k(x, y) \Big] - \mathbb{E}_{x \sim \mathcal{D}}\big[ V_{m_0}^k(x) \big] \tag{27}$$

$$= J_k(\pi_m^k) - J_k(\pi_{m_0}^k), \tag{28}$$

where the last equality uses $J_k(\pi_{m_0}^k) = \mathbb{E}_{x \sim \mathcal{D}}[V_{m_0}^k(x)]$ by definition of $V_{m_0}^k$. Rearranging yields

$$J_k(\pi_m^k) = J_k(\pi_{m_0}^k) + \mathbb{E}_{x \sim \mathcal{D}}\Big[ \sum_{y \in \mathcal{Y}_k} \pi_m^k(y | x) \, A_{m_0}^k(x, y) \Big], \tag{29}$$

which is exactly Lemma 4.1. $\qquad\square$

## C.4. Proof of Lemma 4.2 (Expansion Bridge)

Let $m_{k-1}$ be the deployed memory at stage $k-1$, inducing $\pi_{m_{k-1}}^{k-1}$ on $\mathcal{Y}_{k-1}$. Consider its conservative lift $\pi_{m_{k-1}}^{\uparrow}$ on $\mathcal{Y}_k$ as in (20). Then

$$J_k(\pi_{m_{k-1}}^{\uparrow}) = \mathbb{E}_{x \sim \mathcal{D}}\Big[ \sum_{y \in \mathcal{Y}_k} \pi_{m_{k-1}}^{\uparrow}(y | x) \, r_k(x, y) \Big] \tag{30}$$

$$= \mathbb{E}_{x \sim \mathcal{D}}\Big[ \sum_{y \in \mathcal{Y}_{k-1}} \pi_{m_{k-1}}^{k-1}(y | x) \, r_k(x, y) \Big], \tag{31}$$

where the second line uses that $\pi_{m_{k-1}}^{\uparrow}(y | x) = 0$ for all $y \in \mathcal{Y}_k \setminus \mathcal{Y}_{k-1}$. By Assumption C.1, for all $y \in \mathcal{Y}_{k-1}$ we have $r_k(x, y) = r_{k-1}(x, y)$, hence

$$J_k(\pi_{m_{k-1}}^{\uparrow}) = \mathbb{E}_{x \sim \mathcal{D}}\Big[ \sum_{y \in \mathcal{Y}_{k-1}} \pi_{m_{k-1}}^{k-1}(y | x) \, r_{k-1}(x, y) \Big] = J_{k-1}(\pi_{m_{k-1}}^{k-1}), \tag{32}$$

proving Lemma 4.2. $\qquad\square$

## C.5. Proof of Theorem 4.3 (Monotonicity Across Expansions)

Fix any $k \geq 1$. By Assumption C.2, the conservative fallback memory $m_{k-1}^{\uparrow}$ is included in the feasible edit set $\mathcal{U}_k(m_{k-1})$. Moreover, it trivially satisfies the KL constraint in (16) since

$$\bar{D}_{\mathrm{KL}}\big( \pi_{m_{k-1}^{\uparrow}}^k \,\|\, \pi_{m_{k-1}^{\uparrow}}^k \big) = 0 \leq \delta. \tag{33}$$

Therefore, $m_{k-1}^{\uparrow}$ is a feasible candidate for the optimization (16). Let $m_k$ be any optimal solution returned by (16). Optimality implies

$$\mathcal{L}_{m_{k-1}^{\uparrow}}^k(m_k) \geq \mathcal{L}_{m_{k-1}^{\uparrow}}^k(m_{k-1}^{\uparrow}). \tag{34}$$

Applying Lemma 4.1 with baseline $m_0 = m_{k-1}^{\uparrow}$ gives, for any $m$,

$$\mathcal{L}_{m_{k-1}^{\uparrow}}^{k}(m) = J_k(\pi_m^k). \tag{35}$$

Substituting into (34) yields

$$J_k(\pi_{m_k}^k) \geq J_k(\pi_{m_{k-1}^{\uparrow}}^k). \tag{36}$$

By Assumption C.2 (realizability of the lifted policy), $\pi_{m_{k-1}^{\uparrow}}^k \equiv \pi_{m_{k-1}}^{\uparrow}$. Thus (36) becomes

$$J_k(\pi_{m_k}^k) \geq J_k(\pi_{m_{k-1}}^{\uparrow}). \tag{37}$$

Finally, by Lemma 4.2, $J_k(\pi_{m_{k-1}}^{\uparrow}) = J_{k-1}(\pi_{m_{k-1}}^{k-1})$. Combining the last two displays gives

$$J_k(\pi_{m_k}^k) \geq J_{k-1}(\pi_{m_{k-1}}^{k-1}), \tag{38}$$

which proves Theorem 4.3. $\qquad\square$

*Table 3.* GAIA comprehensive results for convenient reference in the appendix. This table reproduces Table 1 and additionally reports scale-up results (w/o memory updates) for DeepSeek-V3.2 and Gemini-3-Pro at larger agent pool sizes ($N \in \{5, 7, 10\}$).

| MODEL CONFIGURATION | LEVEL 1 | LEVEL 2 | LEVEL 3 | OVERALL |
|---|---|---|---|---|
| *Open-Source Baselines* | | | | |
| DEEPSEEK-V3.2 (3 AGENTS) | 64.15% | 45.35% | 46.15% | 51.52% |
| 5 AGENTS (W/O MEMORY) | 64.15% | 56.98% | 34.62% | 55.76% ↑+8.2% |
| 7 AGENTS (W/O MEMORY) | 56.60% | 56.98% | 42.31% | 54.55% ↑+5.9% |
| 10 AGENTS (W/O MEMORY) | 50.94% | 52.33% | 34.62% | 49.09% ↓-4.7% |
| QWEN3-235B-INSTRUCT (3 AGENTS) | 58.49% | 47.67% | 26.92% | 47.88% |
| GLM-4.7 (3 AGENTS) | 62.26% | 54.65% | 42.31% | 55.15% |
| *Proprietary Baselines* | | | | |
| GPT-5 (3 AGENTS) | 60.38% | 54.65% | 42.31% | 54.55% |
| GPT-5-HIGH (3 AGENTS) | 67.92% | 41.86% | 23.08% | 47.27% |
| GPT-5.2 (3 AGENTS) | 66.04% | 36.05% | 46.15% | 47.27% |
| GEMINI-2.5-PRO (3 AGENTS) | 60.38% | 53.49% | 46.15% | 54.55% |
| GEMINI-3-PRO (3 AGENTS) | **69.81%** | **62.79%** | **34.62%** | **60.61%** |
| 5 AGENTS (W/O MEMORY) | 73.58% | 67.44% | 42.31% | 65.45% ↑+8.0% |
| 7 AGENTS (W/O MEMORY) | 56.60% | 54.65% | 38.46% | 52.73% ↓-13.0% |
| 10 AGENTS (W/O MEMORY) | 67.92% | 60.47% | 26.92% | 57.58% ↓-5.0% |
| *Naive Scale-up (Qwen-3-30B-A3B-Instruct, w/o Memory)* | | | | |
| 3 AGENTS | 49.06% | 47.67% | 26.92% | 44.84% |
| 5 AGENTS | 54.72% | 43.02% | 23.08% | 43.64% |
| 7 AGENTS | 54.72% | 40.70% | 46.15% | 46.06% |
| 10 AGENTS | 47.17% | 44.19% | 26.92% | 42.42% |
| *MonoScale (Qwen-3-30B-A3B-Instruct, w/ Memory)* | | | | |
| 3 AGENTS | 49.06% | 47.67% | 26.92% | 44.84% |
| 5 AGENTS | 52.83% | 45.35% | 42.31% | 47.27% ↑+8.3% |
| 7 AGENTS | 60.38% | 47.67% | 30.77% | 49.09% ↑+6.6% |
| 10 AGENTS | 60.38% | 55.81% | 42.31% | 55.15% ↑+30.0% |

# D. Additional Results and Analysis

## D.1. GAIA Comprehensive Results

**Analysis.** Table 3 further shows that naive scale-up without memory updates is not reliably safe, even for strong routers in an ideal agent pool where all workers function normally. While DeepSeek-V3.2 and Gemini-3-Pro can obtain short-term gains when expanding from 3 to 5 agents (e.g., overall accuracy improves by +8.2% and +8.0%, respectively), the trend is non-monotonic: performance can regress substantially as more agents are added (e.g., Gemini-3-Pro drops by $-13.0\%$ at 7 agents), suggesting that a larger pool can amplify mis-routing to unsuitable workers. In contrast, MonoScale improves consistently as $N$ increases, supporting our claim that scaling requires an explicit update mechanism to stabilize routing decisions and prevent collapse in an expanding, heterogeneous agent pool.

## D.2. Ablation Study

To isolate the contribution of MonoScale's key design choices, we ablate two components on GAIA with the Qwen3-30B-A3B router: (i) *negative memory*—the negative routing constraints distilled from *failed* warm-up traces—and (ii) *customized warm-up*—the agent-conditioned task synthesis. For the non-customized variant, we follow the cold-start setting of EvoRoute (Zhang et al., 2026a) and use 50 standardized TaskCraft (Shi et al., 2025) tasks as generic warm-up data. As shown in Table 4, both components matter, especially at larger scales. Removing negative memory causes a sharp drop at 10 agents ($55.2\% \rightarrow 40.0\%$), indicating that explicit "when *not* to route" constraints are essential for stable expansion. Replacing customized warm-up with generic tasks also lowers the final expanded accuracy ($55.2\% \rightarrow 48.5\%$). Overall, MonoScale's gains stem from both negative routing memory and deployment-aware, agent-conditioned onboarding, rather than from additional warm-up or memory volume alone.

*Table 4.* Ablation study on GAIA (overall accuracy, Qwen3-30B-A3B router). *w/o Negative Memory* removes the negative routing constraints distilled from failed traces; *w/o Customized Warm-up* replaces agent-conditioned task synthesis with generic TaskCraft tasks. Both components contribute, most visibly at the largest agent pool.

| CONFIGURATION | 5 AGENTS | 7 AGENTS | 10 AGENTS |
|---|---|---|---|
| MONOSCALE (FULL) | 47.3% | 49.1% | **55.2%** |
| W/O NEGATIVE MEMORY | 45.5% | 50.9% | 40.0% |
| W/O CUSTOMIZED WARM-UP | 50.9% | 49.7% | 48.5% |

*Table 5.* Cross-router memory transfer on GAIA (overall accuracy). *w/ native memory*: the Gemini-3-Flash router uses memory it generated itself. *w/ Qwen memory*: the Gemini-3-Flash router directly reuses the memory *generated by the Qwen-3-30B-A3B-Instruct router*, with no adaptation. The 3-agent column is the shared base pool before any expansion, so the two settings coincide there; transfer effects appear only from the first expansion ($N \geq 5$).

| SETTING | 3 AGENTS | 5 AGENTS | 7 AGENTS | 10 AGENTS |
|---|---|---|---|---|
| GEMINI-3-FLASH, W/ NATIVE MEMORY | 49.1% | 49.7% | 60.6% | 62.4% |
| GEMINI-3-FLASH, W/ QWEN MEMORY | 49.1% | 53.9% | 60.0% | 58.8% |

### D.3. Cross-Router Memory Transfer

A natural question is whether the routing memory accumulated by MonoScale is tied to the specific router that produced it, or whether it captures router-agnostic knowledge about the agent pool. To probe this, we take the memory *generated by the Qwen-3-30B-A3B-Instruct router* during expansion and apply it *directly, without any adaptation*, to a different router backbone, Gemini-3-Flash, on GAIA across the same $3 \rightarrow 10$ agent expansion. Table 5 compares this transferred setting (*w/ Qwen memory*) against Gemini-3-Flash running MonoScale with its own natively generated memory (*w/ native memory*).

Two observations follow. First, MonoScale remains effective on a frontier, closed-weight router: with native memory, Gemini-3-Flash improves consistently from $49.1\%$ to $62.4\%$ as the pool grows. Second, and more importantly, *memory generated by the Qwen router transfers to Gemini-3-Flash without any adaptation*: the transferred setting still scales from $49.1\%$ to $58.8\%$ with no collapse, and its gap to native memory stays modest and bounded (e.g., $62.4\%$ vs. $58.8\%$ at 10 agents) rather than escalating with scale. This indicates that the memory MonoScale distills encodes transferable, router-agnostic profiles of agent capabilities and failure modes, rather than router-specific artifacts.

### D.4. Scaling Beyond Ten Agents

Our main experiments scale the agent pool up to $N = 10$. To probe behaviour at a larger scale, we further expand the HLE-MCQ system from 10 to 20 worker agents by adding ten Gemini-2.5-Pro-based workers, keeping the rest of the setup unchanged. Overall accuracy improves from $19.88\%$ at 10 agents to $23.4\%$ at 20 agents, indicating that at moderate scale, additional model capability and skill diversity still yield tangible gains and that the context window is not yet the dominant bottleneck. We nonetheless do not claim that the current prompt-based routing paradigm alone addresses true Web-scale settings: at hundreds or thousands of agents, context length becomes the primary bottleneck, and MonoScale would need to be combined with an agent-recommendation or hierarchical-scheduling stage that first retrieves a small relevant subset of agents before routing.

### D.5. Onboarding Cost

Onboarding a new agent is a one-time cost. In our implementation, it requires roughly 50 customized warm-up tasks with 4 rollout trajectories each, costing about \$19.5 per agent under DeepSeek's official API pricing. For comparison, a single full evaluation costs about \$50, and the full HLE benchmark about \$100 (roughly $5\times$ the per-agent onboarding fee). The warm-up procedure is highly parallelizable and typically completes within $10$–$15$ minutes, so its wall-clock latency is limited. Because onboarding is performed only once per agent while the resulting agent is reused across many downstream tasks, we regard this overhead as moderate and readily amortized given the persistent gain in system capability.

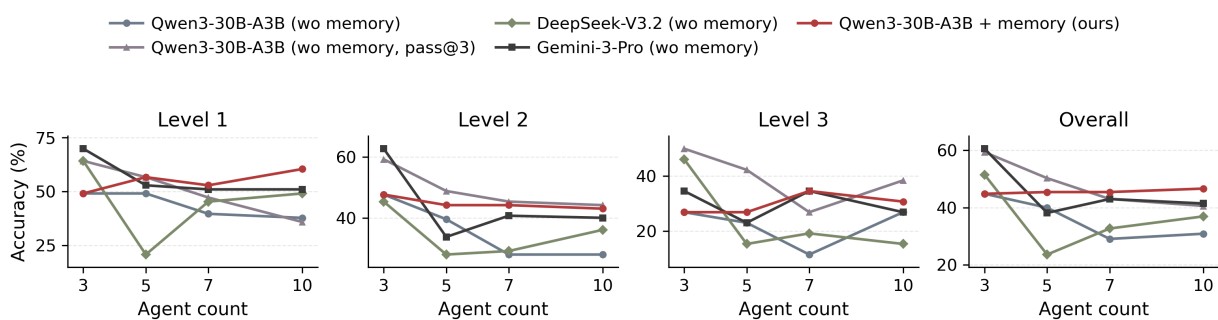

*Figure 5.* Additional results on the noisy/malfunctioning agent pool in GAIA. We report level-wise (L1–L3) and overall accuracy as the agent pool scales ($N = 3, 5, 7, 10$) for multiple routers and inference variants, including a best-of-3 aggregation (`pass@3`) for the Qwen3 (w/o memory) router.

### D.6. Multi-Seed Variance

Full multi-seed evaluation of an LLM-based MAS is expensive, so we assess variance on a randomly sampled 40-task GAIA subset using three independent seeds. The standard deviation of overall accuracy is 2.89, 1.44, and 2.89 points at 5, 7, and 10 agents respectively—only about 0.5–1.2 tasks out of 40. This low variance indicates that MonoScale's gains across expansion are not an artifact of random fluctuation; accordingly, we frame the empirical trend as a stable, non-decreasing improvement rather than a strict per-point monotonicity claim.

### D.7. Error Analysis of the Local Scaling Dip

The GAIA scaling curve in Figure 3 rises in overall trend but is not strictly monotonic at every point: accuracy shows a small local dip at $N = 6$ (47.3% at 5 agents versus 46.1% at 6, a difference of about 1.2 points, or 2 tasks out of the 165-question GAIA validation set). Inspecting this dip, we attribute it predominantly to *tool-budget exhaustion*. After a new agent is integrated and warm-up completes, the increased diversity of the agent pool leads the router to decompose tasks into finer-grained subtasks and distribute them more aggressively across agents. Under a fixed per-task tool-call budget, this finer decomposition raises the number of tool calls a task requires, increasing the chance of hitting the budget limit before a conclusive answer is produced and thus causing the task to fail. These cases are concentrated in complex reasoning tasks that require multi-step sequential tool use, which are inherently most sensitive to budget constraints. We trace the root cause to the warm-up stage: its current task design does not explicitly cover budget-constrained scenarios, so the collected evidence lacks negative signals about budget exhaustion and the distilled routing memory provides little guidance on avoiding over-decomposition under a limited budget. This suggests a concrete remediation within the MonoScale framework—emphasizing budget-constrained scenarios during warm-up so that familiarization captures these failure signals and encodes them into memory. Finally, this local dip does not contradict our analysis: Theorem 4.3 guarantees a non-decrease of *expected* performance across expansion stages, not strict monotonicity of every finite-sample evaluation point.

### D.8. Malfunctioning Agent Pool

**Analysis.** As shown in Fig. 5, when the pool contains only three high-quality worker agents with no obvious deficiencies, routers backed by stronger LMs achieve excellent overall performance on GAIA, substantially outperforming the router using Qwen-3-30B-A3B-Instruct as the backbone. However, as lower-quality workers are introduced, even a router built on the strongest backbone (e.g., Gemini-3-pro) can suffer a sharp performance degradation without warm-up; moreover, Best-of-3 (`pass@3`) does not consistently prevent regressions as the agent pool becomes larger and noisier. This indicates that in noisy, malfunctioning agent pools, zero-shot orchestration can amplify systematic mis-routing to unreliable workers, ultimately causing the MAS to collapse and fail to operate reliably. In contrast, after warm-up with our customized tasks, the router using Qwen-3-30B-A3B-Instruct as the backbone maintains non-decreasing performance even in the same noisy pool, and can even obtain a modest gain as more agents are added. These results underscore the necessity of using MonoScale to warm up the router in real-world Agentic Web settings where noisy agents are prevalent.

# E. MonoScale/Expansion-Aware Memory Schema

## E.1. Overview

This section details the memory schema used in our MonoScale framework to solve the Sequential Augmentation and Cold-Start challenges. Unlike generic memory systems, our memory entries act as the Editable Memory $m$ in the contextual bandit formulation, allowing the Router's policy to be optimized via natural language updates.

These memories are extracted from Expansion Familiarization Traces (specifically, the successes and failures on synthesized warm-up tasks) and stored in JSON format. By capturing auditable and rollbackable routing constraints, the memory system enables safe policy updates that mitigate performance collapse during scale-up.

We establish distinct memory schemas for the **Planning** and **Coordinating** phases. These schemas include both common items—populated with insights distilled from each phase's unique operational perspective—and specific items tailored to their distinct responsibilities.

### E.1.1. KEY COMMON ITEMS

**task_pattern**  Describes when this experience pattern applies, enabling the system to match stored patterns to new incoming tasks. Includes task characteristics (e.g., "requires cross-referencing multiple sources"), required capabilities, and environmental constraints. This drives pattern retrieval during execution.

**actionable_rules**  Concrete do/don't guidance extracted from comparing successful and failed attempts. These rules are prescriptive and directly injected into the Router's prompts to influence decision-making, effectively shaping the policy behavior within safe boundaries.

**specific_guidance**  The most detailed and structured item, containing phase-specific strategies. For **Planning**, includes agent selection logic (which agents to use/avoid) and decomposition strategies. For **Coordinating**, includes agent matching rules and error handling patterns. This is where the core learning resides.

## E.2. Planning Phase Memory

### E.2.1. JSON TEMPLATE

```
{
    "title": "<Pattern name>",
    "task_pattern": "<When this pattern applies>",
    "available_agent_pool": ["<Agent 1>", "<Agent 2>", "..."],
    "success_indicators": [
        "<Observable signal indicating success>",
        "..."
    ],
    "failure_indicators": [
        "<Observable signal indicating failure>",
        "..."
    ],
    "key_insights": [
        "<High-level learning>",
        "..."
    ],
    "actionable_rules": {
        "do": ["<Recommended action>", "..."],
        "dont": ["<Anti-pattern to avoid>", "..."]
    },
    "specific_guidance": {
        "task_characteristics": {
```

```
            "task_type": "<computational | information_gathering | reasoning | hybrid>",
            "complexity_indicators": ["<Indicator>", "..."],
            "required_capabilities": ["<Capability>", "..."]
        },
        "agent_selection": {
            "primary_agents": [
                {
                    "agent": "<Agent name>",
                    "rationale": "<Why choose: strengths, use cases, boundaries>"
                },
                "..."
            ],
            "agents_to_avoid": [
                {
                    "agent": "<Agent name>",
                    "rationale": "<Why avoid: limitations, failure patterns>"
                },
                "..."
            ],
            "selection_criteria": "<How to disambiguate between agents>"
        },
        "decomposition_strategy": {
            "approach": "<sequential | parallel | hybrid>",
            "rationale": "<Why this approach>",
            "subtask_prompt_guidelines": [
                "<How to phrase subtask>",
                "..."
            ]
        },
        "dependency_handling": "<How to manage subtask dependencies>",
        "verification_strategy": "<Whether/how to add verification>"
    },
    "importance": "<high | medium | low>",
    "confidence": 0.0,
    "source_files": ["<trace_file.md>", "..."],
    "original_task": "<Original task question>"
}
```

### E.2.2. PLANNING-SPECIFIC ITEMS

**agent_selection** The core of Planning learning, addressing agent misassignment as the pool scales. Contains:

- primary_agents: Recommended agents with rationales covering strengths, use cases, and capability boundaries. Example: "Web Agent excels at retail searches on official domains; use for current pricing; boundary: avoid third-party aggregators."

- agents_to_avoid: Agents to exclude with rationales on limitations and failure patterns. This field facilitates the **conservative fallback** mechanism by explicitly listing scenarios where invoking the new agent is risky and should be avoided to prevent performance degradation.

- selection_criteria: Disambiguation rules when multiple agents seem applicable (e.g., "Prefer Web Agent over Search Expert for retailer websites; Math Agent over Code Agent for deterministic calculations").

**decomposition_strategy** How to break down tasks: approach (sequential/parallel/hybrid), rationale, and subtask prompt templates.

**dependency_handling**   How to manage information flow between subtasks.

**verification_strategy**   Whether and how to add verification subtasks.

### E.3. Coordinating Phase Memory

E.3.1. JSON TEMPLATE

```
{
    "title": "<Pattern name>",
    "task_pattern": "<When this pattern applies>",
    "available_agent_pool": ["<Agent 1>", "<Agent 2>", "..."],
    "success_indicators": [
        "<Observable signal indicating successful coordination>",
        "..."
    ],
    "failure_indicators": [
        "<Observable signal indicating coordination failure>",
        "..."
    ],
    "key_insights": [
        "<High-level learning about coordination>",
        "..."
    ],
    "actionable_rules": {
        "do": ["<Recommended coordination action>", "..."],
        "dont": ["<Coordination anti-pattern>", "..."]
    },
    "specific_guidance": {
        "agent_matching": {
            "matching_strategy": "<Subtask-to-agent mapping logic>",
            "assignment_examples": [
                {
                    "agent": "<Agent name>",
                    "best_suited_for": "<What this agent excels at>",
                    "avoid_for": "<What NOT to assign>"
                },
                "..."
            ],
            "disambiguation_rules": "<How to choose between similar agents>",
            "prompt_transformation": "<How to adapt planner instructions>"
        },
        "execution_flow": {
            "parallel_opportunities": "<Subtasks that can run concurrently>",
            "critical_path": "<Dependencies that must be sequential>"
        },
        "error_handling": {
            "retry_conditions": ["<When to retry>", "..."],
            "retry_modifications": "<How to modify for retry>",
            "max_retries": 0,
            "fallback_strategy": "<Action after max retries>"
        },
        "result_validation": {
```

```
            "quality_checks": ["<Check>", "..."],
            "rejection_criteria": "<When to reject and reassign>"
        },
        "information_passing": {
            "context_propagation": "<How to pass info between agents>",
            "context_filtering": "<Whether/how to filter context>"
        }
    },
    "importance": "<high | medium | low>",
    "confidence": 0.0,
    "source_files": ["<trace_file.md>", "..."],
    "original_task": "<Original task question>"
}
```

### E.3.2. COORDINATING-SPECIFIC ITEMS

**agent_matching**   The core of Coordinating decision-making, addressing precise agent selection during execution. Contains:

- matching_strategy: High-level logic for mapping subtasks to agents.

- assignment_examples: Concrete examples showing what each agent excels at (best_suited_for) and what to avoid assigning (avoid_for). These examples guide real-time assignment decisions.

- disambiguation_rules: Rules for choosing between similar agents when multiple seem applicable.

- prompt_transformation: How to adapt **Planning** instructions to agent-specific formats and capabilities.

**execution_flow**   Manages parallelization: parallel_opportunities (concurrent subtasks) and critical_path (sequential dependencies).

**result_validation**   Quality assurance: quality_checks and rejection_criteria for when to reject results and reassign.

**information_passing**   Context management: context_propagation (how to pass info) and context_filtering (when to filter/summarize).

# F. Case Study Trajectories: The Anatomy of Scaling Failures and Their Cure

This appendix provides detailed execution trajectories to empirically validate the core motivation and effectiveness of our framework. We structure the analysis into two distinct parts:

1. **The Scaling Trap (Cases 1 & 2)**: We first demonstrate *why* naive scaling fails. By comparing a stable 3-Agent baseline against a naive 10-Agent expansion, we expose two fundamental failure modes—**Cold-Start Misjudgment** and **Brittle Workflow Links**—that arise when a Router relies solely on static agent descriptions without an active familiarization phase.

2. **The MonoScale Resolution (Case 3)**: We then demonstrate *how* our method solves these issues. We explicitly compare the Naive 10-Agent system against the **MonoScale 10-Agent system**, showing how retrieved memory constraints effectively guide the router to utilize multimodal capabilities that were previously mismanaged, turning a potential failure into a success.

### F.1. Case 1: The Trap of Cold-Start Misjudgment (Anagram Solving)

In this case comparison (Table 6), we illustrate the **cold-start misjudgment** (previously referred to as capability confusion) inherent in naive scaling. This case study uses **Qwen-3-30B-Instruct** as the router backbone. The task requires generating a valid anagram from a Shakespearean quote, subject to strict constraints: (1) no punctuation, and (2) exact character rearrangement.

The **3-Agent baseline** (left) correctly routes the task. Its planning phase, operating with a smaller scope, interprets the task as a standard extraction and coding problem. The key divergence occurs in the **10-Agent Naive system** (right), where a newly added **Reasoning Agent** is available. The Router, cold-starting on this new agent, erroneously interprets the **Agent Card** description ("specializes in logical deduction and complex reasoning") as implying the capability to solve complex constraint satisfaction problems (CSPs) like anagrams. However, typical LLM-based reasoning agents excel at logical inference but often struggle with precise character-level counting due to tokenization limits.

Unaware of this **capability boundary**, the Naive Router delegates the core analysis to the Reasoning Agent during the coordinating phase. The agent, being unable to perform exact letter counts, simply **guesses the most famous quote** as the answer, leading to a failure. This highlights a critical failure mode of naive scaling: without an active warm-up phase to expose this limitation, adding the "smarter" Reasoning Agent **can actually degrade system performance**.

*Table 6.* **Case Study 1: Naive Scaling Failure (Backbone: Qwen-3-30B-Instruct).** Comparison between the baseline (3 Agents) and the scaled system (10 Agents). The Router cold-starts on the "Reasoning Agent" and misjudges its ability to handle character-level constraints. The agent fails to count letters and guesses the answer.

| Step | 3-Agent Baseline (Success) | 10-Agent Naive System (Failure) |
|---|---|---|
| 1 | **Router (Plan)**: Decomposes into: (1) Transcribe; (2) Identify/clean phrase; (3) Solve Anagram. | **Router (Plan)**: Decomposes into: (1) Transcribe; (2) Analyze rules and solve. |
| 2 | **Router (Coord)**: Assigns Subtask 1 to **Doc Agent**. | **Router (Coord)**: Assigns Subtask 1 to **Multimedia Agent** (New). |
| 3 | **Worker**: Calls ask_question_about_audio. → Output: "...queries on two fronts about how life turns rotten." | **Worker**: Calls audio2text. → Output: "...queries on two fronts about how life turns rotten." |
| 4 | **Router (Coord)**: Assigns Subtask 2 to **Doc Agent**. | **Router (Coord)**: Assigns Subtask 2 to **Reasoning Agent**. *(Selected due to "Logical Deduction" description)* |
| 5 | **Worker**: Calls execute_code (Regex). → Removes punctuation. Length ≈ 100 chars. | **Worker**: Calls ask_strong_reasoning_llm. → **Error:** Cannot perform exact letter counting. Guesses the most famous quote "To be or not to be" directly. |
| 6 | **Router (Coord)**: Assigns Subtask 3 to **Reasoning Coding Agent**. | |
| 7 | **Worker**: Calls execute_code. → Checks letter freq. Matches "To be or not to be..." | |
| 8 | **Final Answer**: "to be or not to be that is the question..." (Correct). | **Final Answer**: "to be or not to be that is the question" (Incorrect; guess failed to match character count). |

## F.2. Case 2: The Trap of Brittle Workflow Links (Bird Species ID)

This case (Table 7) demonstrates the **brittle workflow links** (previously referred to as context fragmentation) and interface mismatches that arise during naive expansion. Crucially, we use **Gemini-3-Pro** as the router backbone for this case. We aim to highlight that even with a SOTA model as the router, naive expansion **may likely still result in performance collapse** due to lack of grounded execution experience.

While the **3-Agent baseline** succeeds via a generalist pattern (implicitly retaining context within a single Doc Agent), the **10-Agent Naive system** attempts to route the downloading subtask to a newly added specialized **Multimedia Agent**. The core failure stems from the Router's ignorance of the Multimedia Agent's **interface boundaries**. Specifically, while the agent utilizes a general-purpose `download_video` tool, it lacks the specialized authentication headers required to bypass YouTube's anti-bot protections (resulting in a `403 Forbidden` error).

The Gemini Router, assuming the specialist is strictly superior for all video tasks, aggressively routes the URL to it. Crucially, due to strict schema enforcement for the specialized agent, the rich descriptive text from the search result is discarded, passing only the raw URL. When the download fails, the agent lacks the metadata context to recover. This creates a brittle dependency step: the Router blindly trusts the "Specialist" label without knowing the specific failure mode (YouTube → 403 Error), illustrating that **specialization likely creates single points of failure** when the Router lacks experience-based constraints.

*Table 7.* **Case Study 2: Brittle Workflow Links (Backbone: Gemini-3-Pro).** Comparison between Baseline and Naive Scale-up. Even with a strong Gemini router, the naive expansion fails because the Router blindly delegates to a specialist (Multimedia Agent) without knowing its specific limitation (YouTube 403 Error), while also losing text context during the rigid handover.

| Step | 3-Agent Baseline (Success) | 10-Agent Naive System (Failure) |
|:---:|---|---|
| 1 | **Router (Plan)**: Decomposes into: (1) Search video & identify species. | **Router (Plan)**: Decomposes into: (1) Get video URL; (2) Download & analyze video. |
| 2 | **Router (Coord)**: Assigns Subtask 1 to **Doc Agent** (Generalist). | **Router (Coord)**: Assigns Subtask 1 to **Search Agent**. |
| 3 | **Worker**: Calls google_search.
→ Output: URL + Page Snippet ("...Rockhopper Penguins..."). | **Worker**: Calls google_search.
→ Output: URL only (youtube.com/...). (Schema restricted). |
| 4 | **Context**: Implicitly Retained. Agent memory holds description. | **Router (Coord)**: Assigns Subtask 2 to **Multimedia Agent**.
→ **Handover Error:** Passes URL only; drops text context. |
| 5 | **Worker**: Verifies info using retained context. No complex tool needed. | **Worker**: Calls download_video.
→ **Error:** 403 Forbidden / Anti-bot. |
| 6 | **Router (Plan)**: Verifies results. | **Worker**: Lacks fallback context. Resorts to hallucination. |
| 7 | **Final Answer**: "Rockhopper Penguins." (Correct). | **Final Answer**: "Emperor Penguins." (Hallucination). |

## F.3. Case 3: The MonoScale Resolution (Scientific Discovery)

In this final case (Table 8), we demonstrate *how* our method resolves the scaling traps identified above. The task requires distinguishing a "Subject" (Diamond) from a "Probe" (Silicon Nanocrystals) in a scientific paper—a nuance often lost in text but clear in the paper's figures.

We explicitly compare the **Naive 10-Agent System** (which possesses all necessary tools but mismanages them) against the **MonoScale 10-Agent System** (which is guided by memory).

- **Naive Failure**: The Naive system defaults to a text-centric search pattern. The Router, unaware of the ambiguity, assigns the verification subtask to the Search Agent in the coordinating phase, which merely finds more text snippets repeating the same error (**Confirmation Bias**).

- **MonoScale Success**: During the familiarization (warm-up) phase, the Router identified that the newly integrated **Image Agent** is capable of processing literature and specifically attending to visual figures/charts. Furthermore, the Router learned the necessary coordination patterns to enable effective collaboration between the Image Agent and existing text-based agents. In the trajectory below, the Router's planning phase explicitly retrieves this experience, enabling the system to physically ground the verification in visual data (analyzing Figure 1) rather than relying solely on text.

This case confirms that effective scaling is not just about the availability of agents, but about the **memory-guided activation** of the right modality at the right time.

*Table 8.* **Case Study 3: MonoScale Resolution.** Comparison of two 10-Agent systems. The Naive system fails due to generic routing. The MonoScale system succeeds because the Router learned the Image Agent's literature-reading capability and collaboration patterns during warm-up.

| Step | Naive 10-Agent System (Failure) | MonoScale 10-Agent System (Success) |
|---|---|---|
| 1 | **Router (Plan)**: Decomposes into standard steps: (1) Identify article; (2) Extract nano-compound; (3) Verify result. *(Lacks figure constraints)* | **Router (Plan)**: **Retrieves Memory:** *"...multimodal tools can be leveraged to analyze figures... to cross-verify..."* Decomposes into: (1) Search; (2) **Analyze SEM & PL Figures**. |
| 2 | **Router (Coord)**: Assigns to **Search Expert Agent**. | **Router (Coord)**: Assigns to **Search Expert Agent**. |
| 3 | **Worker**: Calls search_serper. → Finds article. Reads text: "...investigated using silicon nanocrystals..." | **Worker**: Calls search_serper. → Finds same target article. Applies strict exclusion for "plasmonics". |
| 4 | **Router (Coord)**: Assigns verification to **Search Expert Agent**. *(Defaulting to text-based verification)* | **Router (Coord)**: **Retrieves Memory:** *"The Image Agent can identify images within the paper PDF."* Assigns analysis to **Image Agent**. |
| 5 | **Worker**: Calls scholar_search. → **Confirmation Bias:** Snippets repeat "using SiNCs". Verifies incorrect entity. | **Worker**: **Image Agent** calls ask_question_about_image. → **SEM Fig.1**: Shows "NCD columns" (Diamond) as structure. → **PL Fig.4**: Shows SiNCs are just surface deposits. |
| 6 | **Router (Plan)**: Aggregates result "silicon nanocrystals". | **Router (Plan)**: Concludes "Diamond" is the primary subject based on structural dominance in SEM. |
| 7 | **Final Answer**: "silicon crystals" (Incorrect). | **Final Answer**: "diamond" (Correct). |

