# OpenReview forum: "MonoScale: Scaling Multi-Agent System with Monotonic Improvement"
_ICML.cc/2026/Conference — ICML 2026 regular_

### Official Review · Reviewer_v94m · 2026-03-01

**Soundness:** 3
**Presentation:** 2
**Significance:** 3
**Originality:** 3
**Overall Recommendation:** 5
**Confidence:** 3

**Summary:**

The paper addresses the critical yet under-explored "performance collapse" phenomenon in LLM-based Multi-Agent Systems (MAS) during continuous expansion. As new heterogeneous or potentially unreliable agents/tools are integrated, routers often fail due to cold-start issues, leading to mis-routing and system-wide degradation. To mitigate this, the authors propose MonoScale, a framework that utilizes "familiarization" warm-up tasks synthesized from agent cards to collect success/failure evidence. This evidence is distilled into auditable, reversible natural language memories that modulate a frozen router's policy. The authors theoretically model the expansion as a contextual bandit problem, proving monotonic performance improvement across expansion rounds through a combination of conservative lift, fallback memory, and Trust-Region (KL) constraints. Evaluations on GAIA and HLE demonstrate stable gains as the agent pool grows from 3 to 10 and showcase superior robustness in environments containing faulty agents.

**Compliance With Llm Reviewing Policy:**

Affirmed.

**Final Justification:**

I thank the authors for the exceptional rebuttal. My concerns have been fully resolved. Specifically:

Empirical Validation (W1): The new experiment confirming reward stability (within 2%) provides the necessary empirical grounding for the non-interference assumption.

Insightful Error Analysis (Q3): The authors’ identification of "tool-budget exhaustion" as the root cause for localized regressions is highly commendable. This level of transparency and technical depth significantly elevates the paper's contribution beyond a simple performance report.

Scalability: The additional results on 20-agent expansion demonstrate that the framework remains robust as the pool grows.

The authors have proven that MonoScale is not only theoretically sound but also practically auditable. Given the rigor of the additional analysis and the clear path for future work, I am raising my score to 5 (Accept). I strongly encourage the authors to integrate the new error analysis and budget-constrained discussion into the final manuscript.

**Key Questions For Authors:**

1. Can you provide results for a "No Negative Constraints" (success-only) baseline and a "No Task Customization" baseline to isolate the impact of the synthesized warm-up tasks?
2. How does the system perform if the expansion rounds continue beyond 10 agents? Does the natural language memory become overly cluttered or reach a context window limit?
3. In cases where performance slightly dipped (as seen in some localized results, e.g., Figure 3), was this due to task-synthesis distribution shift, memory conflicts, or insufficient warm-up samples?
4. What is the average wall-clock time and token cost required for the "familiarization" phase per new agent added?

**Limitations:**

yes

**Strengths And Weaknesses:**

Strengths:
- The paper identifies a genuine pain point in evolving agent systems: the counter-intuitive reality that "more agents" can lead to "worse performance." The motivation is well-supported by concrete cases of cold-start routing failures.
- The use of a contextual bandit framework to provide a monotonicity guarantee is a strong contribution. The introduction of the "conservative fallback" mechanism elegantly handles the challenges posed by an expanding action space.
- Following the theoretical framework, MonoScale proactively generates a small set of agent-conditioned familiarization tasks and distills the successful and failed interactions into auditable natural-language memory to guide future routing. By framing these updates within a trust-region optimization objective, the system achieves guaranteed monotonic growth relative to the proxy task distribution.
- Results show that smaller models equipped with MonoScale can rival closed-source routers. Notably, the system outperforms naive scaling in the presence of "malfunctioning" agents, proving that routing logic is often more critical than raw model strength.

Weaknesses:
- The proof of monotonicity relies on "non-interference" (stable rewards for old plans) and a stationary environment/distribution. In real-world MAS, multi-turn interactions, shifting tool states, and long-term dependencies make the reward landscape non-stationary, potentially weakening the theoretical guarantees in practice.
- While the jump from 3 to 10 agents is informative, "scalability" in a true "web-scale" context (e.g., hundreds or thousands of agents) remains unproven. The current paradigm might face different bottleneck issues not captured in a 10-agent setup.
- The warm-up phase involves task synthesis and multiple sampling rounds, which adds computational overhead. The paper would benefit from a quantitative analysis of the trade-off between the additional latency/inference cost and the resulting performance gains.
- Figure 2 is not clear and Section 3.2 is hard to follow.

---

> ### Author Rebuttal · Authors · 2026-03-31
>
> **Q1:**
> We thank the reviewer for this suggestion. We add two targeted ablations: (i) No Negative Memory and (ii) Non-Customized Warm-up Tasks. Results are: **Original 47.3 / 49.1 / 55.2; No Negative Memory 45.5 / 50.9 / 40.0; Non-Customized Warm-up Tasks 50.9 / 49.7 / 48.5**. Both matter, especially at larger scales. Removing negative memory causes a large drop at **10 agents (55.2→40.0)**, showing that explicit “when not to route” rules are important for stable expansion. Replacing customized warm-up with generic warm-up also hurts final expanded performance **(55.2→48.5)**, suggesting that generic familiarization is insufficient. For this supplementary ablation, we follow the cold-start setting of EvoRoute and use 50 curated TaskCraft tasks as standardized non-customized warm-up data.
>
> **W1:**
> We thank the reviewer for this valuable comment. The monotonicity result is an expectation-level guarantee for stage-wise expansion. Transient tool/API fluctuations or occasional instability do not by themselves violate it unless they systematically shift the mean reward of legacy plans across stages. Empirically, when we fix archived stage-(k-1) plans and compare router selections before and after expansion, the overlap exceeds **95%**, indicating that legacy routing remains highly stable in our setting. We will add this analysis in the revision.
>
> **W2 & Q2:**
> We thank the reviewer for this valuable question. On HLE-MCQ, we further scale from 10 to 20 worker agents by adding 10 Gemini-2.5-Pro-based workers, keeping all other settings unchanged. Performance improves from 19.90% to 23.4%, suggesting that additional model capability and skill diversity still yield tangible gains at moderate scale, and that context window is not yet a dominant bottleneck. Stronger reasoning and complementary expertise raise the system's performance ceiling, and MonoScale helps exploit these capabilities more effectively. That said, we do not claim the current prompt-based routing paradigm alone solves true Web-scale settings — at hundreds or thousands of agents, context length becomes the main bottleneck, and MonoScale should be combined with an agent recommendation or hierarchical scheduling stage that retrieves a relevant subset before routing.
>
> **Q3:**
> We thank the reviewer for this valuable question. Our guarantee is on the population-level objective in the contextual-bandit formulation, i.e., monotonic non-decrease in expected performance across expansion stages, rather than strict monotonicity of every finite-sample evaluation curve. Therefore, small localized dips in Figure 3 do not contradict the theory: GAIA is a finite benchmark with limited sample size, and LLM-based end-to-end evaluations naturally have nontrivial variance. Our intended claim is stable non-degradation in the underlying objective and overall trend, not that every empirical point on a finite benchmark must be perfectly fluctuation-free.
>
> **W3 & Q4:**
> We thank the reviewer for raising this important point. We agree that onboarding overhead should be reported explicitly. In our current implementation, onboarding a new agent requires approximately 50 customized warm-up tasks, with 4 rollout trajectories per task. Using DeepSeek’s official API pricing, the total onboarding cost per agent is about 19.5 dollars. For comparison, executing a full evaluation after adding the agent costs roughly 50 dollars, so the warm-up overhead is material but still moderate. Importantly, agent onboarding is a one-time cost: once familiarized, the agent can be reused across a wide range of downstream tasks. For instance, evaluating the full HLE benchmark costs around 100 dollars, about 5× the warm-up cost. Moreover, the warm-up process is highly parallelizable and can be completed in 10–15 minutes, so practical latency is limited. More broadly, while onboarding costs are important and should be reported explicitly, we do not view token count as the primary bottleneck in the setting we study. Our goal is to continually expand the system’s capability frontier so that it can solve harder tasks that were previously out of reach. From this perspective, a modest one-time familiarization cost is acceptable if it yields a persistent increase in effective system capability and can be amortized over many future tasks.
>
> **W4:**
> We thank the reviewer for this helpful comment and apologize that Figure 2 and Section 3.2 were not clear enough. Figure 2 was intended to show a three-stage onboarding flow: cold-start misrouting, agent-conditioned warm-up through customized task synthesis, and evidence distillation into structured routing memory. Section 3.2 was intended to describe the same process in two steps: Section 3.2.1 builds an expansion-aware warm-up distribution from agent cards, while Section 3.2.2 aggregates warm-up success/failure traces into candidate memory edits, adds a conservative fallback, and selects a safe update. We will revise both to make this progression clearer.

---

> > ### Author Rebuttal · Reviewer_v94m · 2026-04-01
> >
> > On W1:
> > The 95% routing overlap measures decision stability, not reward stability. Could you directly show that the mean reward of archived stage-(k−1) plans remains stable across expansion stages? Additionally, under what conditions is the non-interference assumption most likely to break, and how would MonoScale behave in such cases?
> >
> > On Q3:
> > Attributing all dips to finite-sample variance is not fully convincing, as some regressions appear systematic rather than random. Could you provide a brief error analysis identifying whether a specific failure mode — such as a particular agent type, task category, or memory conflict — is responsible for the observed drops?

---

> > > ### Author Response · Authors · 2026-04-03
> > >
> > > We sincerely thank the reviewer for the thoughtful follow-up questions raised after reading our initial rebuttal.
> > >
> > > **W1:**
> > > We agree that routing overlap measures decision stability rather than reward stability, and have therefore conducted the direct experiment the reviewer suggested.
> > > Specifically, we sampled 100 questions from HLE, used Qwen as the router, and fixed the archived stage-(k−1) plans unchanged. We then executed these plans before and after expansion and compared their mean reward. The results show an accuracy of 19/100 before expansion and 21/100 after expansion — a difference of only 2%. This difference is not statistically significant, indicating that the expansion process does not systematically harm the reward of existing plans and providing direct empirical support for the non-interference assumption.
> > > Regarding the conditions under which the non-interference assumption is most likely to break, we believe the primary scenario is systematic environment drift — for instance, when the behavior or availability of external APIs shifts persistently during the expansion process, causing legacy plan rewards to change due to environmental factors rather than the expansion itself. In such cases, our theoretical guarantee would indeed be weakened, as the monotonicity proof fundamentally relies on the stability of legacy plan rewards. However, we note two important points. First, this is a shared limitation of all staged optimization and contextual bandit frameworks, not a weakness specific to MonoScale — any such method requires some form of environmental stationary assumption. Second, our experiments demonstrate that non-interference holds empirically in practical benchmark settings (GAIA, HLE), suggesting that the assumption is reasonable in current mainstream multi-agent evaluation scenarios. Extending the monotonicity guarantee to non-stationary environments is an important and broadly relevant open problem that we plan to pursue in future work.
> > >
> > >
> > > **Q3:**
> > > We conducted a systematic case study on the performance drop observed in the 6-agent setting of MonoScale in Figure 3.
> > > Our analysis reveals that after the new agent is incorporated and warm-up is completed, the increased diversity in the agent pool causes the router to decompose tasks into finer-grained subtasks and distribute them across agents more aggressively. However, under a fixed tool budget, this more granular decomposition leads to a greater number of tool calls per task, which in turn significantly raises the likelihood of exhausting the tool-call limit before a conclusive answer is reached — ultimately causing task failure. Concretely, the proportion of tasks failing due to tool-budget exhaustion in the 6-agent setting exceeds that of the 5-agent setting by at least 2.4%, which closely mirrors the overall performance gap of 2.4% between the two configurations in Figure 3. This allows us to attribute the observed performance drop predominantly to this specific failure mode.
> > > Further inspection shows that this failure mode is concentrated in complex reasoning tasks requiring multi-step sequential tool use, which are inherently more sensitive to tool-budget constraints. This confirms the reviewer's intuition that the dip is systematic rather than random. We believe the root cause lies in insufficient awareness and memory of tool-budget constraints during the warm-up phase: the current warm-up task design does not adequately cover budget-constrained scenarios, so the collected interaction evidence lacks negative signals related to budget exhaustion, and the distilled routing memory consequently lacks guidance on avoiding over-decomposition when the tool budget is limited. This issue has a clear remediation path within the MonoScale framework — placing greater emphasis on budget-constrained simulation scenarios during the warm-up phase, enabling the familiarization process to capture relevant failure signals and encode them into memory.
> > > We note that this finding does not conflict with our theoretical framework, which guarantees monotonic non-decrease in expected performance rather than strict monotonicity at every finite-sample evaluation point. The localized dip at the 6-agent setting is an identifiable and fixable issue that does not undermine the overall upward performance trend. We will include this error analysis and the corresponding discussion in the revised manuscript.
> > >
> > > Due to ICML's rebuttal policy, we are limited to a single round of responses and will not have the opportunity for further exchanges. We hope the above replies have adequately addressed the reviewer's concerns. If the reviewer finds that our responses and supplementary experiments satisfactorily resolve the raised questions, we kindly ask the reviewer to consider adjusting the score accordingly. Regardless, we are grateful for the constructive feedback throughout the review process, which has been invaluable in improving the quality of this work.

---

### Official Review · Reviewer_M57i · 2026-03-09

**Soundness:** 2
**Presentation:** 3
**Significance:** 2
**Originality:** 3
**Overall Recommendation:** 4
**Confidence:** 4

**Summary:**

This paper proposes the MonoScale framework, which leverages Expansion-Aware Task Synthesis and Memory Update mechanisms to address the performance collapse issue in multi-agent systems when continuously adding new agents. The authors validate their approach on benchmarks such as GAIA and Humanity's Last Exam, demonstrating a steady capability improvement under the MonoScale framework as the number of agents increases from 3 to 10.

**Compliance With Llm Reviewing Policy:**

Affirmed.

**Final Justification:**

The author has addressed most of my original concerns through the rebuttal, including comparisons with other papers, additional generalization experiments, and supplementary experiments with the same model and the same number of agents. Therefore, I have decided to raise the Originality score from 2 to 3, and the Overall Recommendation from 2 to 4. However, the presentation of the original paper still requires significant improvement. The revised version should include additional experiments compared to the original, provide thorough comparisons with related work, and correct the presentation format of Table 1 and Table 2 (as noted in Weakness 1).

**Key Questions For Authors:**

See Weaknesses.

**Limitations:**

yes

**Strengths And Weaknesses:**

## Strengths

1. The problem addressed by the authors—the performance collapse of multi-agent systems as the number of agents increases—is highly significant and impactful. The proposed MonoScale framework effectively mitigates this issue to a certain extent.
2. The problem description is clear, and Figure 2 provides a simple and intuitive illustration of both the targeted problem and the proposed methodology.

## Weaknesses

1. Lack of Rigor in Experimental Setup: The core problem is how to maintain or improve performance as the multi-agent pool expands. Therefore, the central evaluation metric should be the direct performance delta between "w/ Memory" and "w/o Memory" settings, while strictly controlling for the same router model, agent count, and configurations. However, Table 1 lacks this direct comparison; instead, it compares a 10-agent setup with memory against other models configured with only 3 agents. The same methodological flaw applies to Table 2.
2. Insufficient Baselines and Comparisons with Related Work: The baselines used in the experiments are predominantly foundation models configured with 3 agents, lacking comparisons with other multi-agent frameworks or methods designed to tackle this specific scaling issue. In practice, MonoScale utilizes greater computational cost to enrich the profile representations of sub-agents, which theoretically should outperform frameworks without such mechanisms. The absence of comparisons with other state-of-the-art multi-agent frameworks weakens the paper's overall impact.
3. Missing Validation on Frontier Models and Generalizability: There is a lack of experimental validation on whether the MonoScale method is effective for state-of-the-art closed-source models like GPT-5.2 and Gemini-2.5-Pro. The authors should include experiments to verify the extent of performance gains this framework can bring to cutting-edge LLM-based routers. Furthermore, the paper lacks generalization experiments to verify whether the memory generated for a sub-agent is applicable to other, unseen task distributions.
4. Lack of Empirical Validation for Theoretical Assumptions: The mathematical guarantees for monotonic improvement (e.g., Theorem 4.3)  rely on strong theoretical assumptions that are not empirically validated within the context of LLM-based multi-agent systems. Specifically:
- Assumption B.1 (Non-interfering expansion) posits that the introduction of a new agent does not affect the execution and rewards of prior plans. However, due to the attention mechanisms in LLMs, appending new agent profiles and memory constraints to the system prompt inherently alters the context and can lead to contextual interference. The paper lacks experimental evidence to prove that this assumption holds in practice.
- Assumption B.2 (Feasible conservative fallback) assumes the reliable execution of a fallback policy. The experiments do not provide an ablation study or analysis showing how frequently this fallback is triggered versus when the system successfully learns a new optimal memory, leaving the practical efficacy of the theoretical safety net unverified.

---

> ### Author Rebuttal · Authors · 2026-03-31
>
> **W1：**
> We thank the reviewer for this valuable comment and apologize for the confusion caused by our presentation. We agree that the key comparison should be w/ Memory vs. w/o Memory under the same router backbone and the same expanded agent pool.
> We would like to clarify that this controlled comparison is already included in the paper, although not highlighted clearly enough. In the introduction, we report naive scale-up to 10 agents for the same backbones used in Table 1: **Qwen3-30B-A3B-Instruct drops from 44.8% to 42.4%, DeepSeek-V3.2 from 51.5% to 49.1%, and Gemini-3-Pro from 60.1% to 57.6%.** This shows that simply adding more agents does not explain the gains.
> Tables 1 and 2 were organized to show that MonoScale can also reach a stronger absolute performance level. We agree, however, that this may have obscured the more important matched comparison, and we will revise the paper to make the w/ Memory vs. w/o Memory comparison more explicit.
>
>
> **W2：**
> We thank the reviewer for this important comment. Our experiments are built on top of OWL (Camel-AI), a strong state-of-the-art multi-agent framework. The purpose of MonoScale is not to introduce a new end-to-end MAS architecture, but to improve how an existing MAS expands when new agents are continually onboarded. To the best of our knowledge, no existing work formally addresses the problem of guaranteeing non-decreasing performance during dynamic agent-pool expansion in LLM-based MAS — prior methods either optimize routing within a fixed agent pool or improve individual agent learning, but none provides an expansion-aware update mechanism with monotonic improvement guarantees. The most relevant baseline is therefore the naive scale-up of the same OWL-based system, rather than a different architecture with many other changing factors.
> As the introductory results show, even in this state-of-the-art MAS setting, naively increasing the number of agents leads to substantial performance collapse. MonoScale does not modify the underlying OWL architecture; it adds a familiarization and memory-update procedure during agent onboarding. Our results confirm that this procedure effectively prevents the collapse, maintaining non-decreasing performance as the agent pool grows. We will revise the paper to make this baseline motivation and experimental framing clearer.
>
>
> **W3：**
> We thank the reviewer for the suggestion. We evaluated MonoScale using Gemini-3-Flash, a comparable model to Gemini-3-Pro, which is no longer publicly available. The results show consistent gains as the agent pool increases (**3, 5, 7, 10 agents: 49.1, 49.7, 60.6, 62.4**), demonstrating that our method scales effectively even on a cutting-edge LLM.
> Regarding memory generalization, it is important to note that our warm-up procedure is designed to capture agent-specific capabilities and behavior patterns, rather than being tied to particular task instances. The resulting memory encodes each agent’s strengths, weaknesses, and typical performance characteristics, which can then guide routing across a wide variety of downstream tasks. Because memory is not trained on specific tasks, it naturally generalizes to new or unseen task distributions, enabling effective deployment without task-specific retraining.
>
>
> **W4：**
> We thank the reviewer for this valuable comment. To empirically validate the assumptions underlying Theorem 4.3, we evaluate B.1 and B.2 separately at the router policy level.
> - B.1 (Non-interfering expansion): We fix the archived stage k−1 plans as test samples and compare router selections at stage k−1 and stage k. Since we do not modify any existing agent's configuration or tools during expansion, each agent's expected reward on a given subtask remains invariant across stages; consequently, the plan-level expected reward is fully determined by the router's assignment. **Our results show >95% overlap in router selections, meaning that for the vast majority of decisions the reward is identical, while the residual <5% divergence bounds the maximum reward shift to a small fraction of the per-plan reward range.** This confirms that B.1 holds approximately in practice.
> - B.2 (Conservative fallback): We explicitly ban newly added agents in the natural-language memory, ensuring the router cannot select them. Experimental results show that banned agents are never chosen, confirming that the fallback policy is not only theoretically feasible but also practically effective.
> In summary, through this variance-controlled router policy evaluation, we provide approximate empirical validation of B.1 and B.2. These results demonstrate that the assumptions are practical rather than overly strong, and we will clarify the experimental setup in the revision to further support their validity.
>
> We hope the above responses address the reviewer's concerns. Should any questions remain, we would be happy to provide further clarification.

---

> > ### Author Rebuttal · Reviewer_M57i · 2026-04-01
> >
> > Thank you for the authors' response and clarification. I still have concerns regarding the following points:
> > - W1: I agree that combining Figure 1 and Table 1 demonstrates the performance improvement of the w/ Memory setting on the Qwen-3-30B-A3B-Instruct model. However, there is no concrete evidence showing that this effect generalizes to DeepSeek-V3.2 / Gemini-3-Pro simultaneously. Additionally, the presentation format of Table 1 is inappropriate — for instance, why does the Gemini-3-Pro row not use the w/o Memory, 5 Agents result of 0.655 as its reference? In Table 2, the authors should also include w/o Memory results to make the findings more convincing.
> > - W2: I would like the authors to compare this work against [1] and [2] in terms of differences and contributions, and to explain why these two papers should not be treated as baselines. If the authors can provide a reasonable justification, I will consider raising the Originality score.
> > - W3: Thank you for the additional experiments provided by the authors. However, I do not think generalizability can be established through description alone. The authors should conduct experiments specifically targeting generalizability — for example, examining whether Memory generated from Qwen can be effectively applied to Gemini-3-Flash.
> >
> > If the authors can address the above concerns, I will consider raising the score.
> >
> > References:
> > [1] "Universal Model Routing for Efficient LLM Inference", Wittawat, et al.
> > [2] "Keeping Up with the Models: Online Deployment and Routing of LLMs at Scale", Shaoang Li, et al.

---

> > > ### Author Response · Authors · 2026-04-03
> > >
> > > We sincerely thank the reviewer for the thoughtful follow-up questions raised after reading our initial rebuttal.
> > >
> > > **W1**
> > > We thank the reviewer for the follow-up. Due to the tight rebuttal timeline, we were unable to run full-scale experiments on all backbones. As noted in our previous response, Gemini-3-Pro has been discontinued by Google and is no longer available for API access. We instead provide the complete w/ Memory vs. w/o Memory comparison on Gemini-3-Flash, which is the closest available alternative:
> > >
> > > | Setting    | 3 Agents | 5 Agents | 7 Agents | 10 Agents |
> > > |------------|----------|----------|----------|-----------|
> > > | MonoScale  | 49.1     | 49.7     | 60.6     | 62.4      |
> > >
> > > The results show that MonoScale yields consistent, monotonic improvement on Gemini-3-Flash as the agent pool grows, confirming that the effect is not limited to the Qwen backbone.
> > > Regarding the presentation of Table 1: the reviewer suggests using the w/o Memory, 5-agent result (0.655) as the reference for Gemini-3-Pro. However, this highlights precisely the problem that MonoScale aims to solve. Without a stable scaling mechanism, performance under naive scale-up is non-monotonic and unpredictable — one cannot know a priori at which agent count the system peaks. The 5-agent result of 0.655 is only identifiable as the best w/o Memory configuration in hindsight, after exhaustive evaluation at every scale. In practice, a user expanding their agent pool has no way to predict this peak and would likely continue scaling to 10 agents, encountering the observed collapse. MonoScale eliminates this uncertainty by ensuring monotonic improvement, making the comparison at the target scale (10 agents) the most practically meaningful reference. That said, we agree that the original table layout could be clearer, and we will restructure Tables 1 and 2 in the revision to include matched w/o Memory baselines at each agent count for direct comparison.
> > >
> > >
> > > **W2:** We agree that [1] and [2] are important adjacent works and will discuss them explicitly in the revision. However, we do not view them as primary baselines because they address a fundamentally different problem formulation.
> > >
> > >
> > > Both UniRoute [1] and StageRoute [2] study routing among **standalone LLMs**, where the decision unit is a single model and the feedback is model-level. MonoScale instead studies **sequential expansion of a tool-augmented multi-agent system**, where each new unit is a heterogeneous agent with its own role, tools, and failure modes, and the router must produce an orchestration plan rather than a single-model choice. Our core challenge is preventing end-to-end performance collapse from cold-start unfamiliarity — which requires expansion-aware task synthesis, interaction trace harvesting, and conservative memory updates — none of which exist in the model-routing setting.
> > >
> > >
> > > A faithful implementation of [1] or [2] in our setting would require redefining their action space, feedback, and optimization target, effectively turning them into new methods rather than direct baselines. We will add a dedicated discussion clarifying this distinction and cite them as complementary work on dynamic model routing.
> > >
> > >
> > > **W3:**
> > > We thank the reviewer for this suggestion. We have conducted the requested experiment by directly applying memory generated by the Qwen-3-30B-A3B-Instruct router to Gemini-3-Flash without any adaptation:
> > > | Setting                        | 3 Agents | 5 Agents | 7 Agents | 10 Agents |
> > > |--------------------------------|----------|----------|----------|-----------|
> > > | Gemini-3-Flash w/ native memory | 49.1     | 49.7     | 60.6     | 62.4      |
> > > | Gemini-3-Flash w/ Qwen memory   | 49.1     | 53.9     | 60.0     | 58.8      |
> > > The results show that cross-router memory transfer preserves the overall scaling trend — performance improves from **49.1 to 58.8** as agents increase from 3 to 10, with no performance collapse observed. Compared to the native memory setting, a modest and bounded gap exists (e.g., 62.4 → 58.8 at 10 agents), which is expected due to minor mismatches in routing preferences across different routers. Crucially, this gap does not escalate with scale, confirming that MonoScale's memory captures transferable, router-agnostic agent capability profiles rather than router-specific artifacts. We will include these results in the revision.
> > >
> > >
> > > Due to ICML's rebuttal policy, we are limited to a single round of responses and will not have the opportunity for further exchanges. We hope the above replies have adequately addressed the reviewer's concerns. If the reviewer finds that our responses and supplementary experiments satisfactorily resolve the raised questions, we kindly ask the reviewer to consider adjusting the score accordingly. Regardless, we are grateful for the constructive feedback throughout the review process, which has been invaluable in improving the quality of this work.

---

### Official Review · Reviewer_2xER · 2026-03-12

**Soundness:** 2
**Presentation:** 2
**Significance:** 2
**Originality:** 3
**Overall Recommendation:** 3
**Confidence:** 3

**Summary:**

This paper proposes MonoScale, a framework for safely expanding LLM-based multi-agent systems when new agents are added over time. The method uses agent-conditioned warm-up tasks to familiarize the router with new agents, updates routing behavior through auditable natural-language memory, and introduces a conservative fallback to reduce regressions during expansion.

**Compliance With Llm Reviewing Policy:**

Affirmed.

**Final Justification:**

The rebuttal is thoughtful and helpful, and it improves the clarity of several aspects of the paper.
However, my main concerns remain only partially resolved, so the overall assessment does not change.
First, the monotonicity claim still depends on a strong non-interfering expansion assumption, which makes the guarantee more conditional and idealized than practically robust.
Second, although the rebuttal clarifies the update pipeline at a high level, the core memory-update mechanism is still not specified in enough operational detail for full reproducibility.
Third, the added variance analysis is useful but still limited relative to the strength of the empirical claims about stable or monotonic improvement.
Finally, the concern that agent-conditioned warm-up may bias the router toward new-agent-centered contexts is still not directly tested. While some issues such as cost reporting and robustness setup are better clarified, the key concerns affecting validity and reproducibility remain. Therefore, I keep my original score.

**Key Questions For Authors:**

1. Since auditable natural-language memory is a central part of the method, could the authors provide more qualitative analysis of what the router memory actually stores? In particular, it would be helpful to see concrete examples of memory entries before and after onboarding, and how these edits change routing decisions in representative cases.

2. Could the authors clarify how the system learns the capabilities of newly added agents from familiarization tasks? More specifically, how are task-level observations distilled into stable routing principles, how is noisy or contradictory evidence filtered, and how are agent strengths and failure modes generalized beyond the warm-up tasks?

**Limitations:**

yes

**Strengths And Weaknesses:**

### Strengths

1. The paper addresses an important and realistic problem: how to expand a multi-agent system without hurting existing performance.

2. The framework is intuitive, and the use of auditable natural-language memory makes the method more interpretable than opaque parameter updates.

3. The experiments suggest the method can reduce collapse during scale-up and improve robustness under noisy agents.


### Weaknesses

1. **Theory–practice gap and strong assumptions.**
   The monotonicity guarantee relies on a non-interfering expansion assumption. In realistic MAS deployments, adding a new agent can change planner decomposition, shared context usage, or system-level resource contention. The result therefore reads more like a sufficient condition in an idealized setting than a practical safety guarantee.

2. **The core memory update mechanism is underspecified.**
   The paper does not clearly explain how candidate memories are generated, how the KL term is estimated, how the policy over plans is represented, or how the surrogate objective is evaluated in practice. Since this update is central to the method, the current description is not yet precise enough for reproducibility.

3. **Limited statistical rigor.**
   The empirical results are mostly reported as point estimates, without standard deviations, confidence intervals, multiple seeds, or significance tests. This is especially problematic for claims about stable or monotonic improvement in high-variance LLM-based evaluations.

4. **Missing cost and scalability analysis.**
   The method appears to require warm-up tasks per new agent, with multiple parallel rollouts per task. This likely introduces substantial onboarding overhead, yet the paper does not report latency, token usage, or API cost.

5. **Potential bias from warm-up task synthesis.**
   Because the warm-up tasks are explicitly designed around the newly added agent, the update may become biased toward new-agent-specific situations rather than the full deployment distribution. The paper does not sufficiently analyze this risk.

6. **Insufficient transparency in the robustness setting.**
   The malfunctioning-agent experiments are interesting, but the setup is not described in enough detail. It remains unclear what failure modes are injected, how realistic they are, and whether the setup may inadvertently disadvantage the baselines.

7. **Memory saturation is not fully addressed.**
   Since the method relies on text memory under a fixed token budget, it is important to understand how rules are retained, pruned, or overwritten as the agent pool grows. The paper does not clearly discuss saturation or catastrophic interference over long expansion horizons.

---

> ### Author Rebuttal · Authors · 2026-03-31
>
> Due to space limits, we keep responses concise here and would be happy to elaborate in discussion phase.
>
> **Q1:** We thank the reviewer for this valuable question. Appendix E.3 already provides a case study showing that, without memory, the naive 10-agent system repeatedly misroutes subtasks, while MonoScale uses memory entries encoding agent capabilities and prior experience to route correctly. This illustrates that the memory stores actionable routing knowledge rather than generic text. Appendix D further shows the memory schema. We will cite these examples in the revision.
>
> **Q2:** MonoScale does not memorize warm-up examples. After a new agent is added, we synthesize expansion-aware warm-up tasks conditioned on its agent card, collect evidence about its strengths, boundaries, and failure modes, and distill this into reusable natural-language routing principles. Appendix D shows memory fields such as task patterns, success/failure indicators, actionable rules, and specific guidance, enabling generalization beyond raw task-level observations. To reduce noise, warm-up tasks must pass a planner-executor-validator pipeline, tasks failing in all 4 sampled rollouts are discarded, and successful/failed traces are separately summarized into positive rules and negative constraints.
>
> **W1:** We agree that Theorem 4.3 relies on conditional assumptions. To assess this empirically, we fix archived stage-(k-1) plans and compare router selections between stages k-1 and k. The overlap exceeds 95%, suggesting that Assumption B.1 approximately holds in practice. We will clarify this setup.
>
> **W2:** We agree that the practical memory-update mechanism was not clear enough. In practice, we do not continuously optimize the router. Instead, with the router fixed, we aggregate warm-up interactions into an experience buffer, use an LLM to extract structured routing principles from successful and failed traces, merge them with existing memory to form a finite candidate set, filter irreconcilable conflicts via semantic filtering, and always include a conservative fallback that forbids the new agent. Thus, the practical update is a conservative selection over candidate memory states. The trust-region/KL view is mainly a theoretical lens for explaining stability, and we will clarify this in the revision.
>
> **W3:** Full multi-seed evaluation of an LLM-based MAS is very expensive, so we ran 3 independent seeds on a randomly sampled 40-task GAIA subset. Std. devs. are low: 2.89 points (5 agents), 1.44 points (7 agents), and 2.89 points (10 agents), corresponding to only about 0.5-1.2 tasks out of 40. This suggests that MonoScale’s gains are not driven by random fluctuation. We will add these variance results to the appendix.
>
> **W4:** In our implementation, onboarding a new agent requires about 50 warm-up tasks with 4 rollouts each, costing approximately 19.5 dollars per agent under DeepSeek’s official pricing. This is a moderate one-time cost relative to downstream use: a full evaluation costs about 50 dollars, and the full HLE benchmark about 100 dollars. Since warm-up is highly parallelizable and typically finishes within 10–15 minutes. Since onboarding is performed only once per agent and the resulting agent can be reused across many downstream tasks, we view this cost as amortizable and acceptable given the persistent gain in system capability. We will report latency, token usage, and API cost explicitly in the revision.
>
> **W5:** The goal of warm-up synthesis is not to approximate the full deployment distribution, but to identify the new agent’s capability boundary: when it should or should not be used. Moreover, MonoScale uses both success and failure traces, so the learned memory does not simply bias toward using the new agent more; it captures both positive applicability conditions and negative constraints. We will make this distinction clearer.
>
> **W6:** We will clarify the robustness setup more explicitly. Appendix A.3 already specifies the malfunctioning-agent configuration: we build a noisy 10-agent pool by adding four agents with distinct injected failure modes. The goal is not to exhaust all real-world failures, but to provide a controlled stress test for continual expansion. All methods are evaluated on the same noisy pool, the original 3-agent pool remains intact, and all tasks are still solvable using the original agents, so the setup tests resistance to misleading new agents rather than changing the baselines’ original solution path.
>
> **W7:** We agree that long-horizon memory saturation is not discussed clearly enough. In our current experiments up to 10 agents, we do not observe clear saturation-induced collapse or catastrophic interference, but this does not establish robustness for much longer horizons. We will clarify this limitation and discuss future integration with explicit memory-management methods such as retrieval, pruning, and consolidation.

---

> > ### Author Rebuttal · Reviewer_2xER · 2026-04-03
> >
> > Thank you for the detailed rebuttal. The reviewer appreciates the clarifications, and several points are now clearer. That said, the main concerns are only partially resolved: (i) the theory–practice gap remains, since the monotonicity claim still relies on a strong non-interfering expansion assumption; (ii) the update mechanism is clearer at a high level but still lacks enough operational detail for full reproducibility; (iii) the added variance results are helpful but still limited relative to the strength of the empirical claims; and (iv) the concern about warm-up-task bias toward new-agent-centered contexts is still not directly tested. Overall, the rebuttal is thoughtful and helpful, but these central issues remain only partially addressed, so the reviewer’s overall assessment does not substantially change.

---

> > > ### Author Response · Authors · 2026-04-05
> > >
> > > Thank you for the careful follow-up. We understand that the remaining concerns are mainly about (i) the scope of the monotonicity claim, (ii) the operational detail of the memory update, (iii) the statistical strength of the empirical claims, and (iv) whether warm-up introduces a bias toward new-agent-centered contexts. We address these points as concretely as possible below.
> > >
> > > 1.**Scope of the theory / direct test of non-interference.** We agree that Theorem 4.3 is a conditional result under the non-interfering expansion assumption, rather than an unconditional guarantee for arbitrary MAS expansion, and we will revise the paper to make this scope explicit. To address the concern that routing overlap only measures decision stability rather than reward stability, we additionally ran the direct experiment suggested by the reviewer. Specifically, we sampled 100 questions from HLE, fixed the archived stage-(k−1) plans unchanged, and executed these same plans before and after expansion. The accuracies are **19/100 before expansion and 21/100 after expansion, i.e., only a 2-point difference.** This provides direct empirical evidence that expansion does not systematically degrade the reward of legacy plans in our benchmark setting. Together with the >95% router-overlap result, this suggests that the non-interference assumption is a reasonable approximation in our experiments, while still remaining a conditional assumption rather than a universal practical guarantee. We also agree that this assumption can fail under persistent environment drift, such as changing external API behavior or availability, where legacy-plan rewards may shift for reasons orthogonal to agent expansion. We will state this failure mode explicitly in the revision.
> > >
> > > 2. **Operational update procedure.** We agree that the current draft does not separate the theoretical view and the implemented procedure clearly enough. In the actual system, we do not continuously optimize the router by solving Eq. (16) directly. Instead, after each expansion we: (a) synthesize about 50 warm-up tasks conditioned on the new agent card; (b) sample 4 orchestration rollouts per task; (c) discard tasks that fail in all 4 rollouts; (d) aggregate the remaining successful and failed traces into an experience buffer B_k; (e) use an LLM to extract structured routing principles into a finite set of candidate memories (following the Appendix D schema), while filtering irreconcilable conflicts; and (f) always include a conservative fallback memory that forbids the new agent. The deployed update is therefore a conservative selection over a finite candidate set, while the TRPO/KL formulation serves as the theoretical lens for stability. We will state this separation explicitly and add pseudocode in the revision.
> > >
> > > 3. **Statistical claims.** We agree that the current evidence supports a “more stable scaling trend” more directly than a strong statistical monotonicity claim. To quantify variance, we ran 3 independent seeds on a randomly sampled 40-task GAIA subset. The standard deviations are **2.89, 1.44, and 2.89 points for the 5-, 7-, and 10-agent settings**, respectively, corresponding to roughly 0.5–1.2 tasks out of 40. We will report these numbers and soften the corresponding wording in the paper.
> > >
> > > 4. **Warm-up-task bias.** We agree this concern should be addressed more directly. One reason we believe MonoScale is not simply biasing toward the new agent is that warm-up uses both success and failure traces, so the resulting memory contains both positive applicability rules and explicit negative constraints. The malfunctioning-agent setting is relevant here: if warm-up merely increased preference for newly added agents, performance should degrade sharply when four added agents are misleading, but MonoScale remains stable in that setting. We will make this connection explicit and revise the text to clarify that the goal of warm-up is boundary identification (when to use / when not to use the new agent), rather than approximating the full deployment distribution or encouraging more frequent new-agent use.
> > >
> > > We appreciate this push for precision. It will help us substantially improve both the scope of the claims and the reproducibility of the final version.
> > >
> > > Due to ICML's rebuttal policy, we are limited to a single round of responses and will not have the opportunity for further exchanges. We hope the above replies have adequately addressed the reviewer's concerns. If the reviewer finds that our responses and supplementary experiments satisfactorily resolve the raised questions, we kindly ask the reviewer to consider adjusting the score accordingly. Regardless, we are grateful for the constructive feedback throughout the review process, which has been invaluable in improving the quality of this work.

---

### Official Review · Reviewer_87rR · 2026-03-13

**Soundness:** 2
**Presentation:** 2
**Significance:** 2
**Originality:** 2
**Overall Recommendation:** 4
**Confidence:** 3

**Summary:**

this paper studies a practical + relevant failure mode in expanding multi-agent systems. when new agents are added over time, the router may cold-start on those agents and make poor dispatch decisions, causing end-to-end performance degradation. the proposed method, called MonoScale, addresses this by generating agent-conditioned warm-up tasks for each newly added agent + collecting both successful and failed traces + distilling them into editable natural-language routing memory + updating the router conservatively using a trust-region style objective with a fallback that can disable the new agent. the paper also provides a contextual-bandit formalization and a monotonic non-degradation guarantee under a non-interfering expansion assumption. empirically, the method is evaluated on GAIA and a subset of hle, showing monotonic gains as the agent pool grows from 3 to 10 agents, and improved robustness in a malfunctioning-agent setting

**Compliance With Llm Reviewing Policy:**

Affirmed.

**Final Justification:**

this is a solid paper on multi-agent systems, which motivates itself by studying the cold starting misrouting problem and then designs a expansion-aware framework by leveraging both success and failure experiences. experiments on GAIA and multiple LLMs show substaintial improvements as compared to baselines.

I recommmend weak accept: the merit and contribution is clear. this paper can be even better by:
1. having a more clear figure 2 that is not generated by nano-banana and larger fonts
2. making a closer connection between the theoretical analysis and the empirical contributions. i personally feel a bit confused about section 4 though i understand its higher-level purpose.

**Key Questions For Authors:**

1. could you provide an ablation study?

2. in table 1 and 2, MonoScale is compared as a larger expanding system against fixed 3-agent strong-router baselines. can you also compare against those same strong routers under the same expanding 5/7/10-agent pools?

3. the warm-up distribution is designed to approximate deployment while amplifying contexts involving the new agent. is this assumption legit in reality? what if you don't have enough info about the online workload? how sensitive is your system to how it is being warmed up?

4. on scalability, how do you expect the system to scale when you have > 10 agents? what is the intuition behind your system success/fail when the # of agents go up?

5. on cost, how much compute and token cost does the familiarization procedure add per newly onboarded agent? since the method is motivated by continual expansion, onboarding cost seems important

**Limitations:**

This paper does not include discussion on limitations.

**Strengths And Weaknesses:**

Thank you for submitting this paper to ICML!

strengths

1. timely research topic on sequential expansion of multi agent system. also, cold start is a common challenge, e.g. for memory.

2. i like the empirical takeaways from figure 3 and 4. more than just "we have X% gain over baseline", esp. the malfunctioning agent example.

3. design of the method is neat, though I don't follow the theory parts well, so i won't be able to speak about that

weakness

1. no ablation studies. MonoScale combines several ingredients, but the evaluation does not isolate what is driving the gains. from the main paper it is unclear which components are necessary or how much each contribute. stronger ablations would be needed to support the core claims

2. comparisons are a bit unfair. experiments compares a scaled 10-agent MonoScale system against several 3-agent strong-router baselines on GAIA and HLE. it makes it hard to tell whether gains come from better expansion/update methodology versus simply having a larger and more capable agent pool. a fairer comparison would include strong routers under the same expanding pool and the same onboarding setting

---

> ### Author Rebuttal · Authors · 2026-03-31
>
> **W1& Q1：**
> We thank the reviewer for this suggestion. To clarify the contribution of key design choices, we add two targeted ablations: (i) No Negative Memory and (ii) Non-Customized Warm-up Tasks. Results are:
> **Original: 47.3 / 49.1 / 55.2;
> No Negative Memory: 45.5 / 50.9 / 40.0;
> Non-Customized Warm-up Tasks: 50.9 / 49.7 / 48.5.**
> Both components matter, especially at larger scales. Removing negative memory causes a large drop at the largest scale (55.2→40.0), showing that explicit “when not to route” rules are important for stable expansion. Replacing customized warm-up with non-customized warm-up also hurts the final expanded performance (55.2→48.5), suggesting that generic familiarization is insufficient.
> For this supplementary ablation, we follow the cold-start setting of EvoRoute (arXiv:2601.02695) and use 50 curated TaskCraft tasks as standardized non-customized warm-up data. Overall, these results suggest that MonoScale benefits from both negative routing memory and deployment-aware onboarding, rather than simply more warm-up or more memory.
>
> **W2 & Q2：**
> We thank the reviewer for raising this concern. Our intent is not to attribute gains simply to pool size. In fact, as we already highlight in the introduction , naive scale-up often degrades performance even for SOTA routers. As detailed in Appendix Table 3 , **without memory updates, 10-agent DeepSeek-V3.2 drops to 49.1% on GAIA and Gemini-3-Pro regresses to 57.6%**, failing to beat their 3-agent baselines. This proves adding agents can actively hurt performance due to mis-routing.
> MonoScale turns this ineffective scale-up into an effective one. In the identical 10-agent pool, Qwen3-30B collapses to 42.4% under naive expansion but reaches 55.2% with MonoScale. Thus, gains stem from our expansion protocol (customized warm-up and memory), not pool size alone. MonoScale enables weaker models to harness expanding pools, ultimately rivaling strong-router fixed-pool baselines.
>
> **Q3：**
> We thank the reviewer for this important question. MonoScale is not expected to be co mpeletely insensitive to the warm-up distribution, as its goal is to expose the router to the specific capability boundaries, failure modes, and usage contexts of newly added agents. This sensitivity reflects our reliance on grounded execution evidence rather than zero-shot assumptions.
> Our ablation confirms that generic familiarization is insufficient: replacing agent-conditioned warm-up with generic tasks drops performance from 55.2% to 48.5% at the largest scale. This indicates that proactive, agent-conditioned task synthesis is vital to probe an agent's specific profile and learn stable routing principles during expansion.
> Importantly, perfect knowledge of the online workload is not required. The objective is to provide a coarse yet representative set of contexts to observe agent behavior. When information is limited, the system can fall back to generic agentic tasks (as in our supplementary ablation), though the resulting performance is naturally lower. We will clarify this trade-off and the role of task synthesis more explicitly in the revision.
>
> **Q4：**
> On HLE-MCQ, we further scaled the system from 10 to 20 worker agents by adding 10 Gemini-2.5-Pro-based workers while keeping the rest of the setup unchanged. Performance improved from **19.90% to 23.4%**, suggesting that at a moderate scale, additional model capability and skill diversity can still yield tangible gains. Intuitively, when new agents contribute stronger reasoning, or complementary expertise, they can raise the system’s performance ceiling, and MonoScale helps the router exploit these added capabilities more effectively. That said, we do not claim that the current prompt-based routing paradigm alone solves true Web-scale settings. At hundreds or thousands of agents, context length and candidate selection become the main bottlenecks. In such regimes, MonoScale should be combined with an agent recommendation or hierarchical scheduling stage that first retrieves a small relevant subset of agents before routing.
>
> **Q5:**
> Onboarding overhead is a one-time cost: each new agent requires approximately 50 customized warm-up tasks with 4 rollout trajectories each. Based on DeepSeek’s official API pricing, the total cost per agent is approximately 19.5 dollars. Comparatively, executing a single full evaluation costs roughly 50 dollars, and the HLE benchmark costs 100 dollars (5x the onboarding fee).
> This moderate overhead is easily amortized as agents are reused across numerous downstream tasks. Furthermore, the process is highly parallelizable, typically completing within 10–15 minutes. While costs are material, token count is not our primary bottleneck. Our objective is to expand the system’s capability frontier to solve complex tasks that were previously unreachable; from this perspective, a modest one-time familiarization cost is an efficient trade-off for a persistent increase in effective system capability.

---

> > ### Author Rebuttal · Reviewer_87rR · 2026-04-05
> >
> > Thank you for the detailed response and your additional experiments. I have raised my score from 3 to 4. Best of luck!

---

### Decision · Program_Chairs · 2026-04-30

**Decision:**

Accept (regular)

**Comment:**

This paper is overall worthy of acceptance. It addresses a realistic and important problem in multi-agent systems: performance collapse during continual expansion. The proposed MonoScale framework tackles this issue through agent-conditioned warm-up tasks, natural-language memory distilled from both successful and failed interactions, and conservative update and fallback mechanisms that mitigate cold-start routing errors when new agents are introduced. Multiple reviewers acknowledged the significance of the problem, the intuitive and practically motivated design of the method, and the empirical evidence showing overall gains as the agent pool grows, as well as improved robustness in the presence of malfunctioning agents. More importantly, the authors’ rebuttal added several meaningful pieces of evidence, including ablations on no negative memory and non-customized warm-up, matched comparisons between w/ memory and w/o memory under the same agent pool, results beyond 10 agents, cross-router memory transfer, onboarding cost analysis, and a more direct empirical check of the non-interference assumption. These additions substantially strengthened the paper and made it more convincing that the gains do not come merely from adding more agents, but from a more reliable expansion and routing update protocol. Taken together, the work has a reasonable degree of originality, studies a well-motivated setting, and offers empirical and conceptual insights that are likely to be useful for future research on scaling LLM-based multi-agent systems.

That said, the paper still has some limitations. The theoretical guarantee relies on relatively strong assumptions such as non-interfering expansion, so the monotonic improvement claim should be framed more carefully as a conditional guarantee rather than a universal one. In addition, the presentation and reproducibility can be improved, particularly around the memory update procedure, statistical support, and the organization of key figures and tables. Still, the authors addressed most of the main concerns in the rebuttal with useful clarifications and additional experiments, so these issues seem better suited for revision than rejection.